# A multifaceted analysis reveals two distinct phases of chloroplast biogenesis during de-etiolation in *Arabidopsis*

**Rosa Pipitone[1], Simona Eicke[2], Barbara Pfister[2], Gaetan Glauser[3], Denis Falconet[4], Clarisse Uwizeye[4], Thibaut Pralon[1], Samuel C Zeeman[2], Felix Kessler[1]\*, Emilie Demarsy[1,5]\***

[1]Plant Physiology Laboratory, University of Neuchâtel, Neuchâtel, Switzerland; [2]Institute of Molecular Plant Biology, Department of Biology, ETH Zurich, Zurich, Switzerland; [3]Neuchâtel Platform of Analytical Chemistry, University of Neuchâtel, Neuchâtel, Switzerland; [4]Université Grenoble Alpes, CNRS, CEA, INRAE, IRIG-DBSCI-LPCV, Grenoble, France; [5]Department of Botany and Plant Biology, University of Geneva, Geneva, Switzerland

**Abstract** Light triggers chloroplast differentiation whereby the etioplast transforms into a photosynthesizing chloroplast and the thylakoid rapidly emerges. However, the sequence of events during chloroplast differentiation remains poorly understood. Using Serial Block Face Scanning Electron Microscopy (SBF-SEM), we generated a series of chloroplast 3D reconstructions during differentiation, revealing chloroplast number and volume and the extent of envelope and thylakoid membrane surfaces. Furthermore, we used quantitative lipid and whole proteome data to complement the (ultra)structural data, providing a time-resolved, multi-dimensional description of chloroplast differentiation. This showed two distinct phases of chloroplast biogenesis: an initial photosynthesis-enabling 'Structure Establishment Phase' followed by a 'Chloroplast Proliferation Phase' during cell expansion. Moreover, these data detail thylakoid membrane expansion during de-etiolation at the seedling level and the relative contribution and differential regulation of proteins and lipids at each developmental stage. Altogether, we establish a roadmap for chloroplast differentiation, a critical process for plant photoautotrophic growth and survival.

**\*For correspondence:**
felix.kessler@unine.ch (FK);
emilie.demarsy@unige.ch (ED)

**Competing interests:** The authors declare that no competing interests exist.

## Introduction

Seedling development relies on successful chloroplast biogenesis, ensuring the transition from heterotrophic to autotrophic growth. Light is a crucial factor for chloroplast differentiation. For seeds that germinate in the light, chloroplasts may differentiate directly from proplastids present in cotyledons. However, as seeds most often germinate underneath soil, seedling development typically begins in darkness and follows a skotomorphogenic program called etiolation, characterized by rapid hypocotyl elongation and etioplast development. Light promotes seedling de-etiolation, which involves a series of morphological changes, such as cotyledon expansion, hypocotyl growth inhibition, and greening, that accompanies the onset of photosynthesis in chloroplasts. During de-etiolation, etioplast–chloroplast transition is thereby rapidly triggered by light following seedling emergence at the soil surface (*Jarvis and López-Juez, 2013*; *Solymosi and Schoefs, 2010*; *Weier and Brown, 1970*). A hallmark of chloroplast differentiation is the biogenesis of thylakoids, a network of internal membranes where the components of the photosynthetic electron transport chain assemble. Thylakoid biogenesis and the onset of photosynthesis rely on the concerted synthesis and coordinated assembly of chlorophylls, lipids, and proteins in both space and time (*Jarvis and López-Juez, 2013*).

The thylakoids harbor the photosynthetic electron transport chain, which is composed of three complexes: photosystem II (PSII), the cytochrome $b_6f$ complex (Cyt $b_6f$), and photosystem I (PSI). Electron transfer between these complexes is facilitated by mobile electron carriers, specifically the low-molecular-weight, membrane-soluble plastoquinone (electron transfer from PSII to Cyt $b_6f$) and the lumenal protein plastocyanin (electron transfer from Cyt $b_6f$ to PSI) (*Eberhard et al., 2008*). Electron transfer leads to successive reduction and oxidation of electron transport chain components. The final reduction step catalyzed by ferredoxin-NADP(+) reductase (FNR) leads to NADPH production. Oxidation of water by PSII and of plastoquinone by Cyt $b_6f$ releases protons into the lumen, generating a proton gradient across the thylakoid membrane that drives the activity of the thylakoid-localized chloroplast ATP synthase complex. Each of the photosynthetic complexes consists of multiple subunits encoded by the plastid or nuclear genome (*Allen et al., 2011*; *Jarvis and López-Juez, 2013*) PSII and PSI have core complexes comprising 25–30 and 15 proteins, respectively (*Amunts and Nelson, 2009*; *Caffarri et al., 2014*). The antenna proteins from the Light Harvesting Complexes (LHC) surround the PSI and PSII core complexes contributing to the formation of supercomplexes. Cyt $b_6f$ is an eight-subunit dimeric complex (*Schöttler et al., 2015*). Each complex of the electron transport chain has a specific dimension, orientation, and location within the thylakoid membrane, occupying a defined surface, and their dimensions have been reported in several studies giving congruent results (*Caffarri et al., 2014*; *Kurisu et al., 2003*; *van Bezouwen et al., 2017*). During de-etiolation, massive protein synthesis is required for assembly of the highly abundant photosynthetic complexes embedded in thylakoids. The photomorphogenic program is controlled by regulation of gene expression at different levels (*Wu, 2014*). Transcriptome analyses have revealed that upon light exposure, up to one-third of Arabidopsis genes are differentially expressed, with 3/5 being upregulated and 2/5 downregulated (*Ma et al., 2001*). Chloroplast proteins encoded by the nuclear genome must be imported from the cytoplasm (*Jarvis and López-Juez, 2013*). The general chloroplast protein import machinery is composed of the multimeric complexes Translocon at the Outer membrane of the Chloroplast (TOC) and Translocon at the Inner membrane of the Chloroplast (TIC), and selective import is based on specific recognition of transit peptide sequences by TOC receptors (*Agne and Kessler, 2010*; *Richardson and Schnell, 2020*).

Reminiscent of their cyanobacterial origin, chloroplast membranes are composed mostly of glycolipids (mono- and di-galactosyldiacylglycerol; MGDG and DGDG) and are poor in phospholipids compared to other membranes in the cell (*Bastien et al., 2016*; *Block et al., 1983*; *Kobayashi, 2016*). Galactolipids comprise a glycerol backbone esterified to contain a single (MGDG) or double (DGDG) galactose units at the *sn*1 position and two fatty acid chains at the *sn*2 and *sn*3 positions. In addition to the number of galactose units at *sn*1, galactolipids also differ by the length and degrees of saturation of the fatty acid chains. In some species, including Arabidopsis, galactolipid synthesis relies on two different pathways, defined as the eukaryotic and prokaryotic pathways depending on the organellar origin of the diacylglycerol precursor. The eukaryotic pathway requires the import of diacyl-glycerol (DAG) synthesized in the endoplasmic reticulum (ER) into the plastids and is referred to as the ER pathway, whereas the prokaryotic pathway is entirely restricted to the plastid (PL) and is referred to as the PL pathway (*Ohlrogge and Browse, 1995*). As signatures, ER pathway-derived galactolipids harbor an 18-carbon chain, whereas PL pathway–derived galactolipids harbor a 16-carbon chain at the *sn*2 position. In addition to constituting the lipid bilayer, galactolipids are integral components of photosystems and thereby contribute to photochemistry and photoprotection (*Aronsson et al., 2008*; *Kobayashi, 2016*). Thylakoids also contain neutral lipids such as chlorophyll, carotenoids, tocopherols, and plastoquinone. These may exist freely or be associated with the photosynthetic complexes, having either a direct role in photosynthesis (chlorophyll, carotenoids, plastoquinone) or participating indirectly in the optimization of light usage and/or mitigation of potentially damaging effects (tocopherols in addition to carotenoids and plastoquinone) (*Hashimoto et al., 2003*; *van Wijk and Kessler, 2017*).

Past studies used conventional electron microscopy to first describe the architecture of the thylakoid membrane network. Based on these 2D observations, researchers proposed that plant thylakoid membranes are organized as single lamellae connected to appressed multi-lamellar regions called grana. How these lamellae are interconnected was revealed only later following the development of 3D electron microscopic techniques (*Staehelin and Paolillo, 2020*). Tremendous technological progress in the field of electron microscopy has been made recently, leading to improved descriptions of chloroplast ultrastructure (*Daum et al., 2010*; *Daum and Kühlbrandt, 2011*). Electron tomography

substantially improved our comprehension of the 3D organization of the thylakoid network in chloroplasts at different developmental stages and in different photosynthetic organisms, including Arabidopsis (*Austin and Staehelin, 2011*; *Liang et al., 2018*), Chlamydomonas (*Engel et al., 2015*), runner bean (*Kowalewska et al., 2016*), and *Phaeodactylum tricornutum* (*Flori et al., 2017*). Electron tomography also provided quantitative information on thylakoid structure such as the thylakoid layer number within the grana stack and the thickness of the stacking repeat distance of grana membrane (*Daum et al., 2010*; *Kirchhoff et al., 2011*). These quantitative data allowed a greater understanding of the spatial organization of the thylakoid membrane in relation to the embedded photosynthetic complexes (*Wietrzynski et al., 2020*). Although electron tomography offers extraordinary resolution at the nanometer level, its main drawback is a limit to the volume of the observation, enabling only a partial 3D reconstruction of a chloroplast. Serial Block Face-Scanning Electron Microscopy (SBF-SEM) is a technique where the embedded specimen is imaged by scanning the face of the block with an electron beam. After imaging, the face of the block is shaved automatically (e.g. 60-nm-thick slices) by an ultramicrotome mounted in the vacuum chamber. The section is discarded and the newly revealed block face is imaged again. Repeated imaging and cutting allows the collection of a tomographic sequence of hundreds of images of the same area. Thereby, a much larger volume can be reconstructed in 3D to show cellular organization (*Peddie and Collinson, 2014*; *Pinali and Kitmitto, 2014*).

In combination with electron microscopy, biochemical fractionation of thylakoids has revealed differential lipid and protein compositions of the grana and the stroma lamellae. The grana are enriched in DGDG and PSII, whereas the stroma lamellae are enriched in MGDG, Cyt *b6/f*, and PSI (*Demé et al., 2014*; *Koochak et al., 2019*; *Tomizioli et al., 2014*; *Wietrzynski et al., 2020*). Changes in lipid and protein compositions during etioplast–chloroplast transition are tightly linked to the thylakoid architecture. In particular, changes in MGDG to DGDG ratio are correlated with the transition from prolamellar body (PLB) and prothylakoid (PT) structures (tubular membrane) to thylakoid membranes (lamellar structure) (*Bottier et al., 2007*; *Demé et al., 2014*; *Mazur et al., 2019*).

Individual studies have provided much insight regarding specific dynamics of the soluble chloroplast proteome, the chloroplast transcriptome, photosynthesis-related protein accumulation and photosynthetic activity, chloroplast lipids, and changes in thylakoid architecture (*Armarego-Marriott et al., 2019*; *Dubreuil et al., 2018*; *Kleffmann et al., 2007*; *Kowalewska et al., 2016*; *Liang et al., 2018*; *Rudowska et al., 2012*). However, these studies were mostly qualitative, focused on one or two aspects, and were performed in different model organisms. Therefore, chemical data related to thylakoid biogenesis remain sparse and quantitative information is rare. Here, we present a systems-level study that integrates quantitative information on ultrastructural changes of the thylakoids with lipid and protein composition during de-etiolation of Arabidopsis seedlings.

## Results

### The photosynthetic machinery is functional after 14 hr of de-etiolation

We analyzed etioplast–chloroplast transition in Arabidopsis seedlings grown in the absence of exogenous sucrose for 3 days in darkness and then exposed to constant white light (*Figure 1A*). These experimental conditions were chosen to avoid effects of exogenous sucrose on seedling development and variations due to circadian rhythm. Upon illumination, the etiolated seedlings switched from the skotomorphogenic to the photomorphogenic developmental program, evidenced by opening of the apical hook and cotyledon greening and expansion (*Figure 1B*; *Kami et al., 2010*). We stopped the analysis following 96 hr of illumination (T96), before the emergence of the primary leaves. Samples were collected at different selected time points during de-etiolation (*Figure 1A*).

In angiosperms, chlorophyll synthesis arrests in the dark but starts immediately upon seedling irradiation (*Von Wettstein et al., 1995*). Chlorophyll levels in whole seedlings increased within the first 4 hr of illumination (T4) and continued to increase linearly during subsequent illumination as the seedlings grew (*Figure 1C*). To evaluate photosynthetic efficiency during de-etiolation, we measured chlorophyll fluorescence and calculated the maximum quantum yield of PSII (Fv/Fm, *Figure 1D* and *Figure 1—figure supplement 1*). PSII maximum quantum yield increased during the initial period of illumination and was near the maximal value of 0.8 at 14 hr of light exposure (T14), independent of light intensity (*Figure 1D* and *Figure 1—figure supplement 1A*). Other photosynthetic parameters

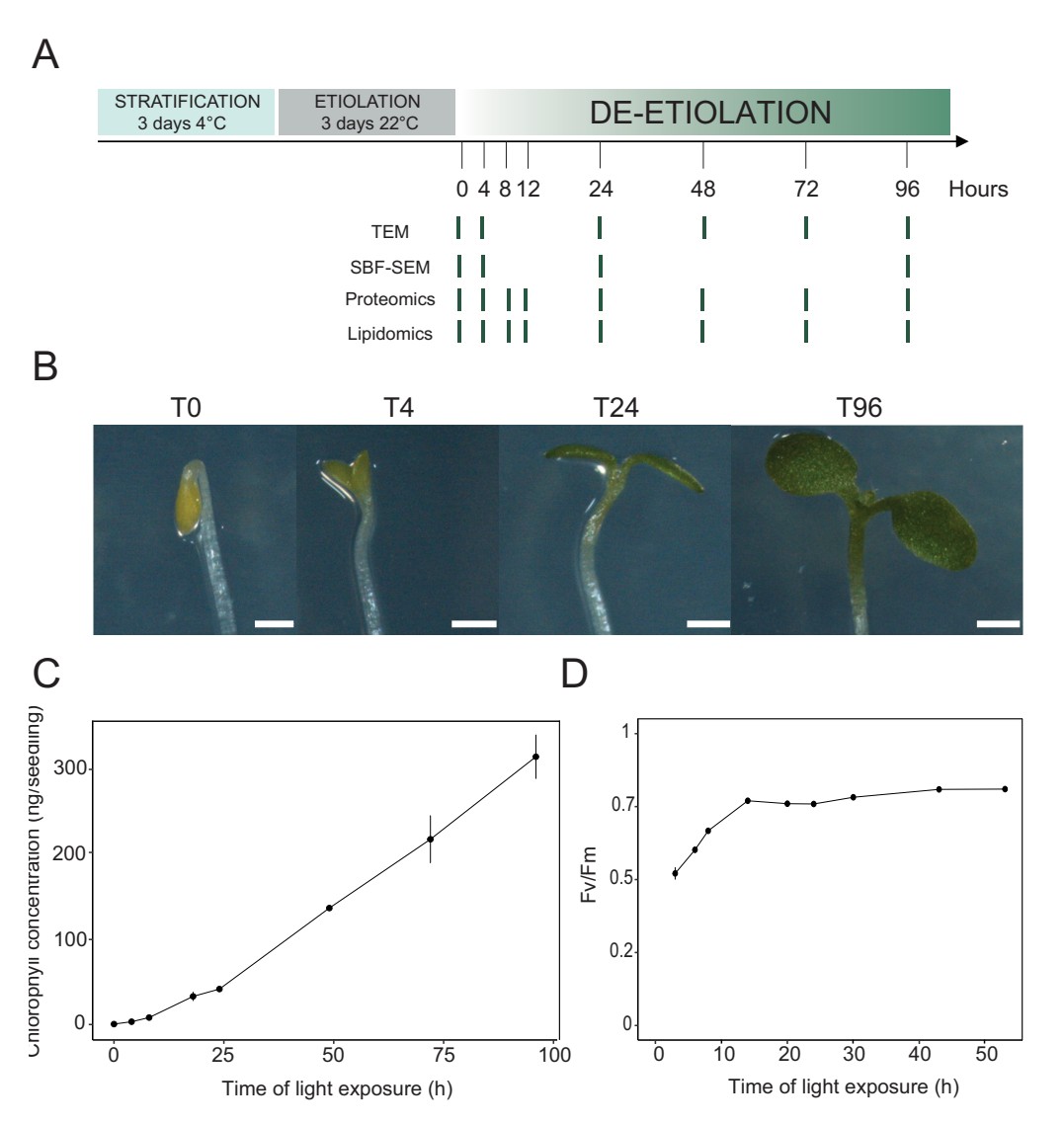

**Figure 1.** Photosynthesis onset during de-etiolation. (**A**) Scheme of the experimental design. Seeds of *Arabidopsis thaliana* (Columbia) sown on agar plates were stratified for three days at 4°C and then transferred to 22°C in the dark. After 3 days, etiolated seedlings were exposed to continuous white light (40 μmol/m$^2$/s) and harvested at different time points during de-etiolation. Selected time points used for different analyses are indicated. (**B**) Cotyledon phenotype of etiolated seedlings (T0) after 4 hr (T4), 24 hr (T24), and 96 (T96) hr in continuous white light. Scale bars: 0.5 mm. (**C**) Chlorophyll quantification at different time points upon illumination. Error bars indicate ± SD (n = 3). (**D**) Maximum quantum yield of photosystem II (Fv/Fm). Error bars indicate ± SD (n = 4–10). For some data points, the error bars are inferior to the size of the symbol. Measurements of further photosynthetic parameters are presented in *Figure 1—figure supplement 1*.

The online version of this article includes the following figure supplement(s) for figure 1:

**Figure supplement 1.** Photosynthesis parameters during de-etiolation.

(photochemical quenching, qP and PSII quantum yield in the light, ΦPSII, *Figure 1—figure supplement 1B and C*) reached maximum values at T14 and remained stable thereafter, indicating that the assembly of a fully functional photosynthetic machinery occurs within the first 14 hr of de-etiolation, and that further biosynthesis of photosynthesis related compounds is efficiently coordinated.

## Major thylakoid structural changes occur within 24 hr of de-etiolation

We determined the dynamics of thylakoid biogenesis during the etioplast–chloroplast transition by observing chloroplast ultrastructure in cotyledons using transmission electron microscopy (TEM)

(*Figure 2*). Plastids present in cotyledons of etiolated seedlings displayed the typical etioplast ultrastructure with a paracrystalline PLB and tubular PTs (*Figure 2A*). The observed PLBs were constituted of hexagonal units with diameters of 0.8–1 μm (*Figure 2E*). By T4, the highly structured PLBs progressively disappeared and thylakoid lamellae were formed (*Figure 2B*). The lamellae were blurry and their thickness varied between 15 and 70 nm (*Figure 2F*). After 24 hr of illumination (T24), the density of lamellae per chloroplast was higher than that at T4 due to an increase in lamellar length and number. Appressed regions corresponding to developing grana stacks also appeared by T24 (*Figure 2C and G*). These early grana stacks consisted of 2–6 lamellae with a thickness of 13 nm each (*Figure 2—figure supplement 1*). In addition, starch granules were present at T24, supporting the notion that these chloroplasts are photosynthetically functional and able to assimilate carbon dioxide ($CO_2$). At T96, thylakoid membrane organization was visually similar to that at T24, but with more layers per grana (up to 10 lamellae per grana; *Figure 2D and H*). In addition, singular lamella thickness at T96 increased by 2–3 nm compared to that at T24 (*Figure 2—figure supplement 1*). The major differences observed between T24 and T96 were increases in starch granule size and number and overall chloroplast size (*Figure 2C and D* and *Table 1*). Etioplast average length (estimated by measuring the maximum distance on individual slices) was 2 μm (±0.9, n = 10) in the dark

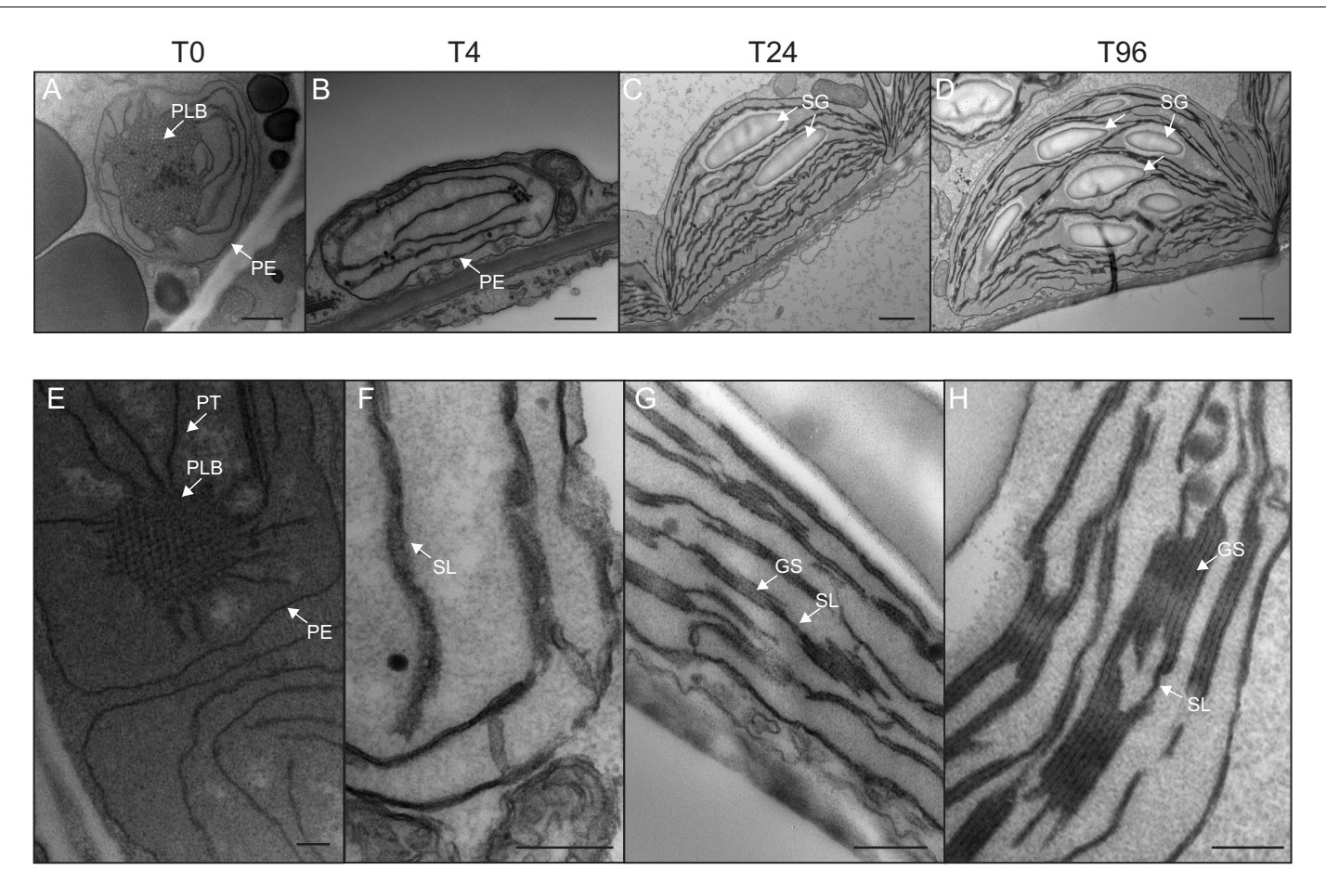

**Figure 2.** Qualitative analysis of chloroplast ultrastructure during de-etiolation. Transmission electron microscopy (TEM) images of cotyledon cells of 3-day-old, dark-grown *Arabidopsis thaliana* (Columbia) seedlings illuminated for 0 hr (T0, A and E), 4 hr (T4, B and F), 24 hr (T24, C and G), and 96 hr (T96, D and H) in continuous white light (40 μmol/m²/s). (A–D) Scale bars: 500 nm, (E–H) higher magnification of A–D images; Scale bars: 200 nm. PLB: prolamellar body; PT: prothylakoid; PE: plastid envelope; SG: starch grain; GS: grana stack; SL: single lamella. Specific details for measurements of lamella thickness are provided in *Figure 2—figure supplement 1*.

The online version of this article includes the following figure supplement(s) for figure 2:

**Figure supplement 1.** Measurement of lamella thickness.

**Table 1.** Collection of quantitative data.

Morphometric data corresponding to thylakoid surfaces and volumes, thylakoid/envelope surface ratio, and chloroplast and cell volumes were collected after SBF-SEM and 3D reconstruction. Chloroplast and cell volumes were also quantified by subsequent confocal microscopy analysis, whereas plastid length was measured using TEM images. Molecular data for galactolipids (GLs) were analyzed by lipidomics, whereas PsbA, PsaC, and PetC were quantified by quantitative immunodetection.

| | *Method* | T0 | T4 | T8 | T12 | T24 | T48 | T72 | T96 |
|---|---|---|---|---|---|---|---|---|---|
| Chloroplast volume (µm³) | SBF-SEM | 12.27 (±2.3) | 9.4 (±4.8) | - | - | 62 (±2.04) | - | - | 112.14 (±4.3) |
| Thylakoid surface (µm²) | SBF-SEM | - | 67 (±29.5) | - | - | 1476 (±146) | - | - | 2086 (±393) |
| Grana lamellae/total thylakoid surface | | - | - | - | - | 2.55 (±0.11) | - | - | 2.08 (±0.57) |
| Thylakoid/envelope surface | | - | 1.02 (±0.15) | - | - | 7.37 (±0.51) | - | - | 6.83 (±1.40) |
| Length of plastid (µm) | TEM | 2 (±0.90) | 2.8 (±0.90) | - | - | 5.1 (±1.47) | - | - | 6 (±1.62) |
| Stroma lamellae volume (µm³) | SBF-SEM | | 2.43 (±0.95) | - | - | 17.87 (±1.04) | - | - | 29.17 (±1.94) |
| Chloroplast volume (µm³) | Confocal | - | - | - | - | 61.5 (±11.2) | 70.1 (±10.2) | 85 (±22) | - |
| Cell volume (µm³) | SBF-SEM | 1173 (±284) | 1891 (±362) | - | - | 6103 (±1309) | - | - | 52597 (±12671) |
| Cell perimeter (µm) | TEM | | | | | 55.3 (±14.1) | 46.4 (±6.1) | 71.7 (±19.1) | 92.8 (±22.1) |
| Number of chloroplast per cell | SBF-SEM | 22 (±6) | 25 (±8) | - | - | 26 (±6) | - | - | 112 (±29) |
| Number of cells per seedling | | - | - | - | - | ~3000 | - | - | ~3000 |
| Protein / GLs surface | | 0.19 (±0.05) | 0.23 (±0.04) | 0.34 (±0.03) | 0.52 (±0.07) | 0.80 (±0.14) | 0.80 (±0.17) | 0.78 (0.07) | 0.87 (±0.25) |
| GLs (nmol/seedling) | Lipidomics | 0.31 (±0.03) | 0.31 (±0.02) | 0.32 (±0.02) | 0.54 (±0.02) | 0.67 (±0.04) | 1.28 (±0.12) | 1.84 (±0.01) | 2.20 (±0.09) |
| PsbA (nmol/seedling) | Immuno-detection | 6.9E-06 (±1.8E-06) | 9.2E-06 (±1.7E-06) | 1.5E-05 (±0.07E-05) | 3.2E-05 (±0.4E-05) | 9.3E-05 (±2E-05) | 2.0E-04 (±0.6E-04) | 3.9E-04 (±0.4E-04) | 6.2E-04 (±1.7E-04) |
| PsaC (nmol/seedling) | Immuno-detection | | | | 1.6E-05 (±0.2E-05) | 7.3E-05 (±2E-05) | 1.1E-04 (±0.7E-04) | 1.7E-04 (±0.4E-04) | 2.3E-04 (±1E-04) |
| PetC (nmol /seedling) | Immuno-detection | 2.7E-05 (±0.8E-05) | 2.8E-05 (±1E-05) | 2.5E-05 (±0.4E-05) | 5.3E-05 (±2.2E-05) | 1.2E-04 (±0.4E-04) | 1.8E-04 (±0.E-04) | 5.7E-04 (±1.8E-04) | 7.9E-04 (±3.7E-04) |

(T0), whereas chloroplast average length was 6 µm (±1.62, n = 10) at T96 (*Table 1*). Collectively, these data show that photosynthetically functional thylakoid membranes form rapidly during the first 24 hr of de-etiolation. This implies that there are efficient mechanisms for thylakoid assembly and structural organization. Subsequent changes seem to involve the expansion of pre-existing structures (i.e. lamellae length and grana size) and the initiation of photosynthetic carbon fixation (reflected by starch content).

## Quantitative analysis of thylakoid surface area per chloroplast during de-etiolation

To visualize entire chloroplasts and thylakoid networks in 3D, and to obtain a quantitative view of the total thylakoid surface area during chloroplast development, we prepared and imaged cotyledons at different developmental stages by SBF-SEM (*Figure 3A–D*). PLBs, thylakoids, and envelope membranes were selected, and segmented images were used for 3D reconstruction (*Figure 3E–L*, and *Videos 1–4*; see also *Figure 2—figure supplement 1* and *Figure 3—figure supplement 1* for grana segmentation). Similar to that observed by TEM (*Figure 2*), a drastic switch from PLB to

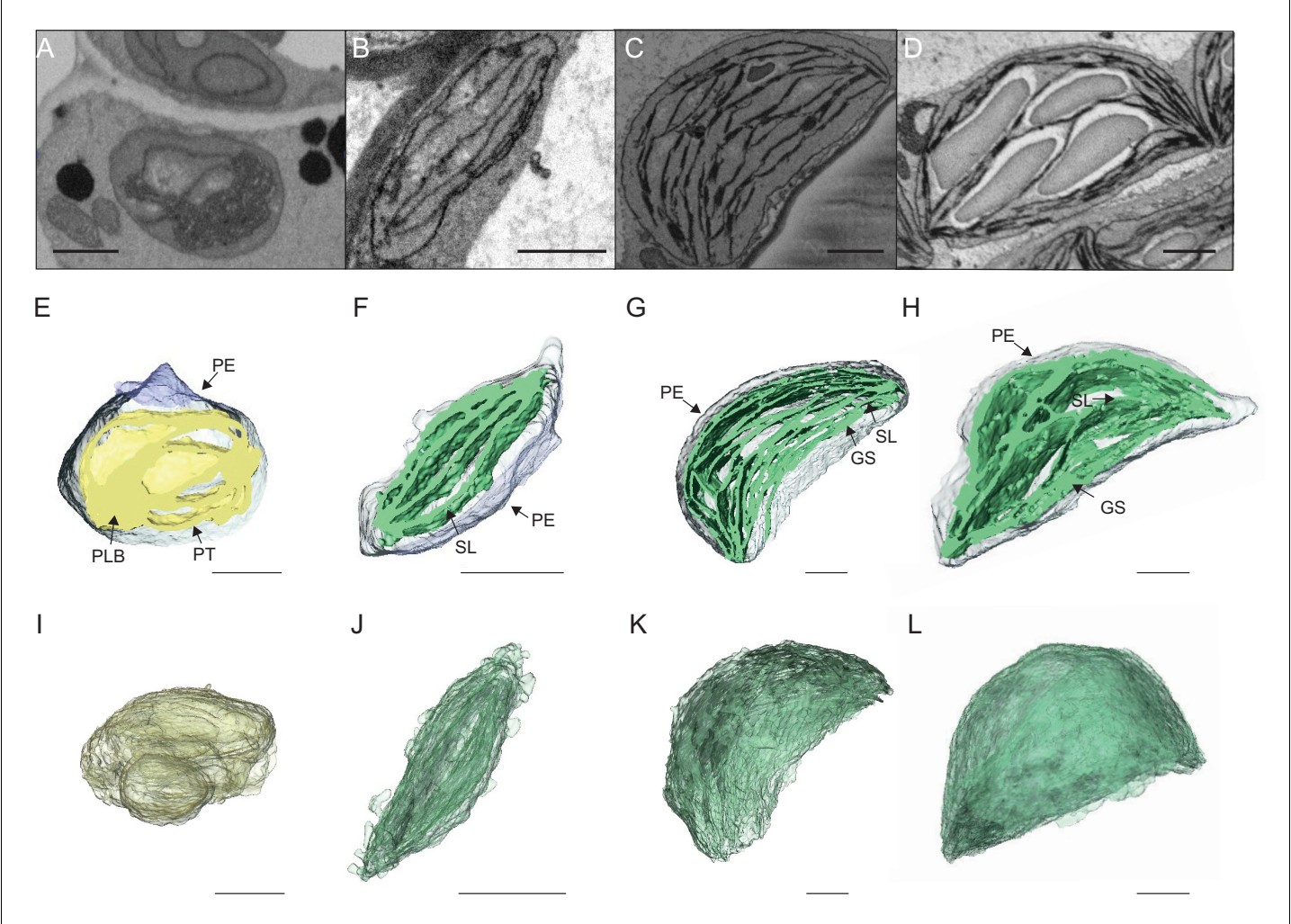

**Figure 3.** 3D reconstructions of chloroplast thylakoid networks during de-etiolation. (A–D) Scanning electron microscopy (SEM) micrographs of representative etioplasts and chloroplasts from 3-day-old, dark-grown *Arabidopsis thaliana* seedlings illuminated for 0 hr (T0; **A**), 4 hr (T4; **B**), 24 hr (T24; **C**), and 96 hr (T96; **D**) in continuous white light (40 µmol/m²/s). (E–H) Partial 3D reconstruction of thylakoid membranes (green) and envelope (blue) at T0 (**E**), T4 (**F**), T24 (**G**), and T96 (**H**). Z-depth of thylakoid membrane reconstruction corresponds to 0.06 µm (**E**), 0.10 µm (**F**), 0.13 µm (**G**), and 0.15 µm (**H**). (I–L) 3D reconstruction of a thylakoid membrane of an etioplast at T0 (**I**) or a chloroplast at T4 (**J**), T24 (**K**), and T96 (**L**). Scale bars = 1 µm. Details of grana segmentation at T24 are provided in **Figure 3—figure supplement 1**. PLB: prolamellar body; PT: prothylakoid; PE: plastid envelope; SG: starch grain; GS: grana stack; SL: single lamella.

The online version of this article includes the following figure supplement(s) for figure 3:

**Figure supplement 1.** Grana segmentation (T24).

thylakoid membrane occurred by T4: the typical structure of the PLB connected to PTs disappeared leaving only elongated lamellar structures (**Figure 3E–F** and **Videos 1** and **2**). At T24 and T96, thylakoid membranes were organized in appressed and non-appressed regions and large spaces occupied by starch granules were observed (**Figure 3G–H** and **Videos 3** and **4**). 3D reconstruction revealed a change in plastid shape from ovoid at T0 and T4 to hemispheric at T24 and T96 (**Figure 3I–L**).

Using 3D reconstruction of the thylakoid network for three or four chloroplasts for each developmental stage, quantitative data such as chloroplast volume and membrane surface area were extracted and calculated (**Figure 4A and B**, **Figure 3—figure supplement 1** and **Table 1**). The total chloroplast volume increased about 11-fold from T4 (9.4 µm³) to T96 (112.14 µm³) (**Table 1**). In parallel, the thylakoid surface area (stroma side) increased about 30-fold reaching 2086 (±393) µm² per chloroplast at T96 (**Figure 4A** and **Table 1**). The surface area increased drastically between T4 and

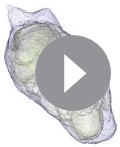

**Video 1.** Representative sequential sections showing etioplasts (T0) followed by segmentation and 3D reconstruction of envelope (blue), and prothylakoids and prolamellar body (yellow) of a single etioplast. The tour of the etioplast reveals its ovoid shape. The sequential view of the 3D reconstruction and final partial 3D visualization reveals a single prolamellar body and interconnected prothylakoids.

https://elifesciences.org/articles/62709#video1

T24 (about 22-fold) and much less (about 1.4-fold) between T24 and T96. Accordingly, quantification of the envelope surface area indicated that the ratio of the thylakoid to envelope surface area increased drastically from T4 to T24, but decreased slightly between T24 and T96 (*Table 1*).

Our quantitative observations confirmed that during chloroplast development the major ultrastructural changes (disappearance of prolamellar body, build-up of the thylakoids and their organization into grana) occurs within the first 24 hr of de-etiolation, and no drastic changes occur thereafter. We further analyzed these temporal processes at the molecular level focusing on proteins and lipids that constitute the thylakoid membrane.

## Dynamics of plastid proteins related to thylakoid biogenesis

We analyzed the full proteome to reveal the dynamics of protein accumulation during de-etiolation. Total proteins were prepared from 3-day-old etiolated seedlings exposed to light for 0–96 hr (eight time points; *Figure 1A*) and quantified by label-free shot-gun mass spectrometry. For relative quantification of protein abundances between different samples, peptide ion abundances were normalized to total protein (see Materials and methods). We considered further only those proteins that were identified with a minimum of two different peptides (with at least one being unique; see Materials and methods for information on protein grouping), resulting in the robust identification and quantification of more than 5000 proteins.

Based on this proteomic approach, the first 12 hr of illumination (T12) saw very few statistically significant changes in protein abundance (*Figure 5—source data 1*). Considering a q-value <0.01 as a stringent threshold value, significant changes were observed only after 8 hr of illumination. These changes correspond to the decreased abundance of only one protein (the photoreceptor cryptochrome 2, consistent with its photolabile property) and increased levels of only three proteins,

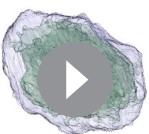

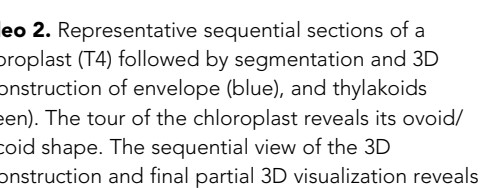

**Video 2.** Representative sequential sections of a chloroplast (T4) followed by segmentation and 3D reconstruction of envelope (blue), and thylakoids (green). The tour of the chloroplast reveals its ovoid/discoid shape. The sequential view of the 3D reconstruction and final partial 3D visualization reveals that thylakoids are constituted by lamellae parallelly oriented.

https://elifesciences.org/articles/62709#video2

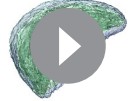

**Video 3.** Representative sequential sections of a chloroplast (T24) followed by segmentation and 3D reconstruction of envelope (blue), and thylakoids (green). The tour of the chloroplast reveals its hemispheric shape. The sequential view of the sections reveals the presence of 8 starch granules . The sequential view of the 3D reconstruction and final partial 3D visualization reveals that thylakoids are constituted by non-appressed (stroma lamellae) and appressed regions (grana).

https://elifesciences.org/articles/62709#video3

which belonged to the chlorophyll a/b binding proteins category involved in photoprotection (AT1G44575 = PsbS; AT4G10340 = Lhcb5; AT1G15820 = Lhcb6; *Chen et al., 2018*; *Li et al., 2000*). Relaxing the statistical threshold value to 0.05, cryptochrome 2 and Lhcb6 levels were respectively decreased and increased already after 4 hr of illumination and the abundance of two other proteins (ATCG00790 = Ribosomal protein L16; AT4G15630 = Uncharacterized protein family UPF0497) increased slightly (fold changes of 1.9 and 1.7, respectively). At 8 hr, a total of 36 proteins displayed a change in abundance with a q-value <0.05. A drastic change of proteome composition occurred by T24, with 402 proteins showing a significant increase in abundance with over two-fold change (FC >2; q-value <0.01) compared with the etiolated stage, and 107 proteins showing a significant decrease with over twofold change (FC <0.5; q-value <0.01). As expected, the 100 most-upregulated proteins comprised proteins related to photosynthesis, proteins constituting the core and antennae of photosystems, and proteins involved in carbon fixation (*Figure 5—source data 1*).

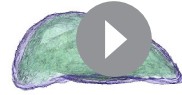

**Video 4.** Representative sequential sections of a chloroplast (T96) followed by segmentation and 3D reconstruction of envelope (blue), and thylakoids (green). The tour of the chloroplast reveals its hemispheric shape. The sequential view of the sections reveals the presence of 11 large starch granules . The sequential view of the 3D reconstruction and final partial 3D visualization reveals that thylakoids are constituted by non-appressed (stroma lamellae) and appressed regions (grana), with large spaces between lamellae occupied by starch granules.
https://elifesciences.org/articles/62709#video4

To monitor the dynamics of the plastidial proteome, we selected proteins predicted to localize to the plastid (consensus localization from SUBA4; *Hooper et al., 2017*). Generation of a global heatmap for each of the 1112 potential plastidial proteins revealed different accumulation patterns (*Figure 5—figure supplement 1* and *Figure 5—figure supplement 1—source data 1*). Hierarchical clustering showed a categorization into six main clusters. Cluster 1 (purple) contained proteins whose relative amounts decreased during de-etiolation. Clusters 2, 5, and 6 (pink, light green, and dark green, respectively) contained proteins whose relative amounts increased during de-etiolation but differed with respect to the amplitude of variations. Proteins in clusters 2 and 6 displayed the largest amplitude of differential accumulation. Gene ontology (GO) analysis (*Mi et al., 2019*) indicated a statistically significant overrepresentation of proteins related to the light reactions of photosynthesis in clusters 2 and 6 (*Figure 5—figure supplement 1—source data 1*). Underrepresentation of organic acid metabolism, in particular carboxylic acid metabolism, characterized cluster 2, whereas overrepresentation of carboxylic acid biosynthesis and underrepresentation of photosynthetic light reactions were clear features of cluster 3. Protein levels in cluster 3 changed only moderately during de-etiolation in contrast with proteins levels in cluster 2. No biological processes were significantly over- or underrepresented in clusters 1, 4, and 5.

To analyze the dynamics of proteins related to thylakoid biogenesis, we selected specific proteins and represented their pattern of accumulation during de-etiolation (*Figure 5*). We included proteins constituting protein complexes located in thylakoids (complexes constituting the electron transport chain and the ATP synthase complex) and proteins involved in chloroplast lipid metabolism, chlorophyll synthesis, and protein import into the chloroplast. In agreement with that depicted in the global heatmap (*Figure 5—figure supplement 1*), all photosynthesis-related proteins increased in abundance during de-etiolation (*Figure 5A*). However, our hierarchical clustering did not show any particular clustering per complex. Only few chloroplast-localized proteins related to lipid biosynthesis were present in our proteomics data set. Among the eight detected proteins, two appeared differentially regulated; fatty acid binding protein 1 (FAB1) and fatty acid desaturase 7 (FAD7) levels increased only between 72 hr of illumination (T72) and T96, whereas the other proteins gradually accumulated over the course of de-etiolation (*Figure 5B*). Etioplasts initiate synthesis of chlorophyll precursors blocked at the level of protochlorophyllide synthesis, with protochlorophyllide oxidoreductase A (PORA) in its inactive form accumulating to high levels in the etioplast before subsequently decreasing at the protein level upon activation and degradation following light exposure

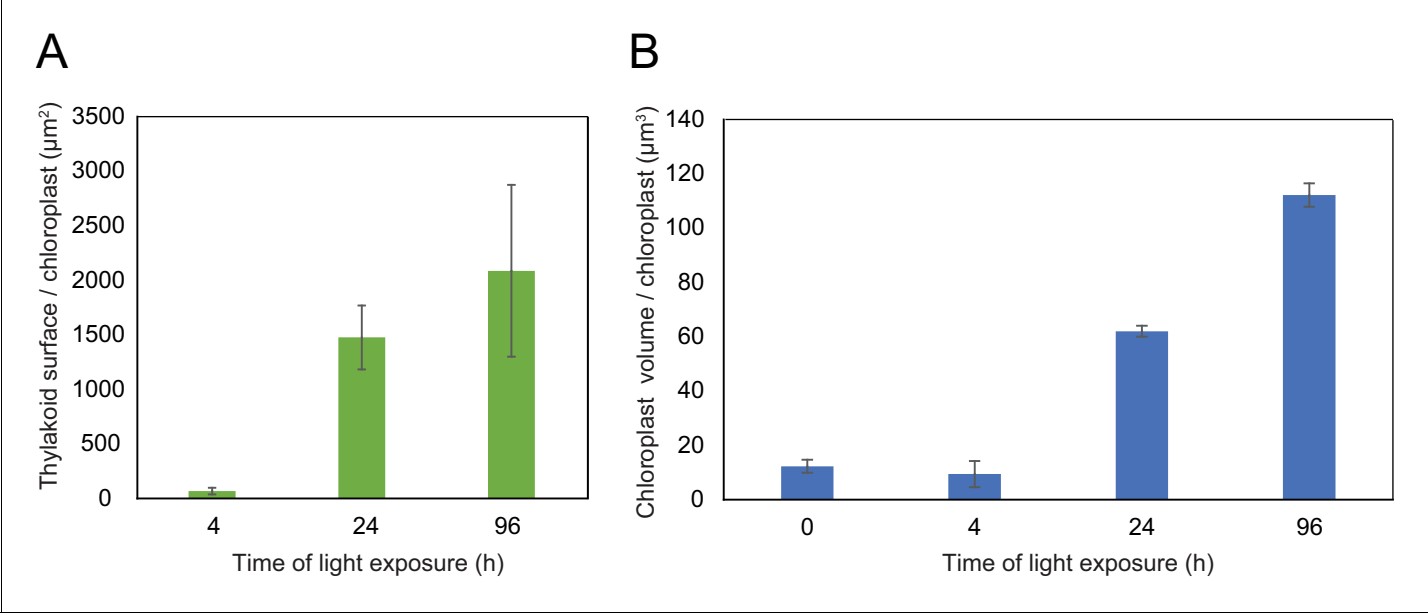

**Figure 4.** Quantitative analysis of chloroplast volume and thylakoid surface during de-etiolation. Quantification of thylakoid surface per chloroplast (**A**) and chloroplast volume (**B**) using 3-day-old, dark-grown *Arabidopsis thaliana* (Columbia) seedlings illuminated for 0 hr, 4 hr, 24 hr, and 96 hr in continuous white light (40 µmol/m$^2$/s). Morphometric data were quantified by Labels analysis module of Amira software. Error bars indicate ± SD (n = 3). The total thylakoid surface indicated in A corresponds to the thylakoid surface exposed to the stroma, calculated in Amira software, in addition to the percentage of the grana surface (%Gs) calculated as described in *Figure 3—figure supplement 1*.

The online version of this article includes the following source data for figure 4:

**Source data 1.** Quantitative chloroplast morphomotric data.

(*Blomqvist et al., 2008*; *Runge et al., 1996*; *Von Wettstein et al., 1995*). In agreement, illumination resulted in increased amounts of most of all detected proteins of the chlorophyll biosynthesis pathway, except PORA and to a lesser extent PORB, which clearly decreased and were separated from other chlorophyll-related proteins (*Figure 5C* and *Figure 5—source data 1*). We also selected proteins involved in protein import in chloroplasts, focusing on the TOC-TIC machinery (*Figure 5D*) that is the major route for plastid protein import and essential for chloroplast biogenesis (*Kessler and Schnell, 2006*). Past studies identified several TOC preprotein receptors that are proposed to display differential specificities for preprotein classes (*Bauer et al., 2000*; *Bischof et al., 2011*). The composition of plastid import complexes varies with developmental stages and in different tissues, thereby adjusting the selectivity of the import apparatus to the demands of the plastid and influencing its proteome composition (*Demarsy et al., 2014*; *Kubis et al., 2003*). Accordingly, the TOC receptors TOC120 and TOC132, which are important for the import of proteins in non-photosynthetic tissues, were more abundant in etioplasts compared to fully-developed chloroplasts (compare T0 and T96). TOC120 and TOC132 were part of a cluster separated from other components of the plastid machinery, such as the TOC159 receptor associated with large-scale import of proteins in chloroplasts. The general import channel TOC75 (TOC75 III) maintained stable expression levels throughout de-etiolation, reflecting its general role in protein import. All other components clustered with TOC159 and displayed gradual increases in accumulation during de-etiolation. Most of these components have not been reported to confer selectivity to the import machinery, which suggests an overall increase of chloroplast protein import capacity.

To validate and complement our proteomic data, we used immunoblot analysis to detect and quantify representative proteins linked to photomorphogenesis and etioplast-to-chloroplast transition.

Our proteomic data indicated a significant decrease of the abundance of the photoreceptor phyA between 48 and 72 hr of illumination (*Figure 5—source data 1*). However, immunoblots revealed that the abundance of phyA dropped already during the first 4 hr of light exposure (*Figure 6*), as previously reported (e.g. *Debrieux and Fankhauser, 2010*). The transcription factor

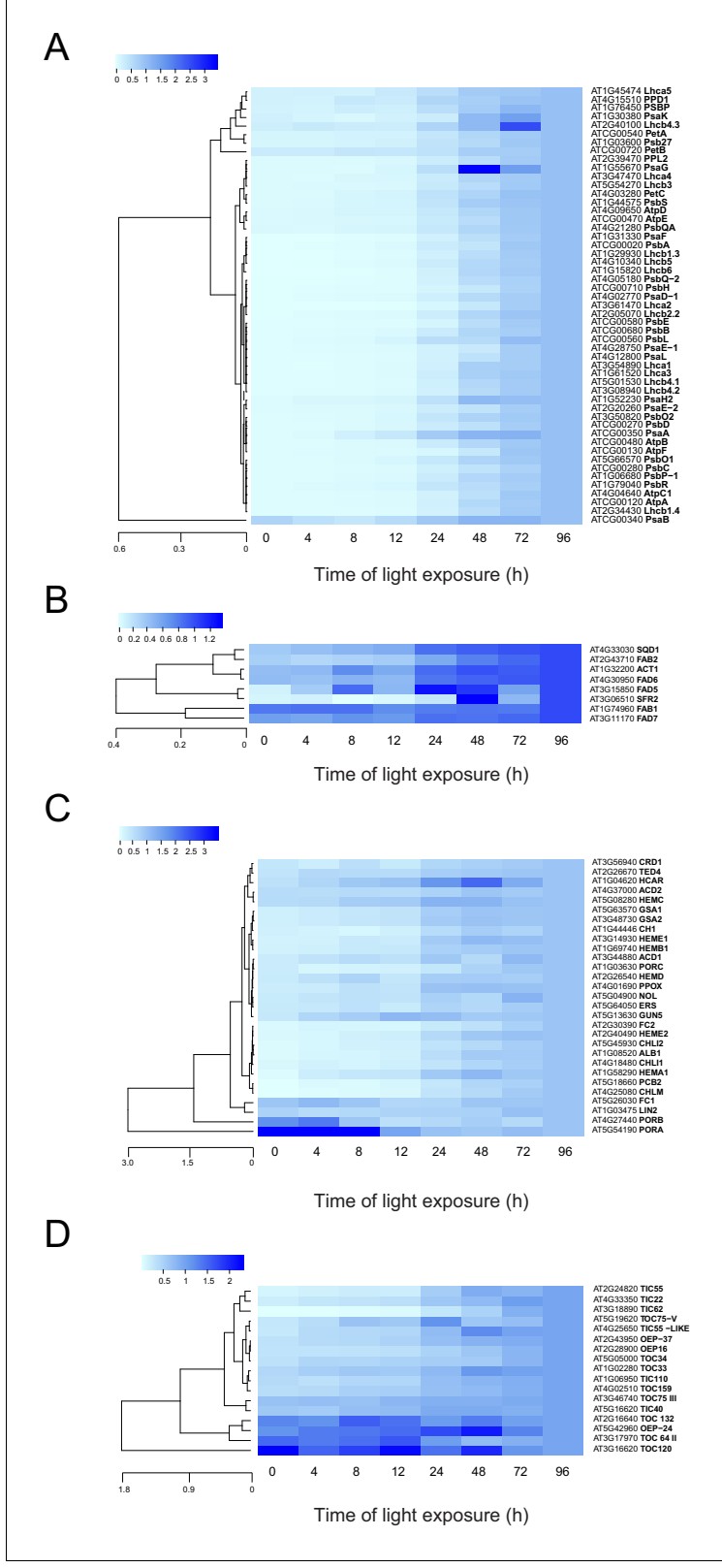

**Figure 5.** Accumulation dynamics of plastid proteins during de-etiolation. Three-day-old etiolated seedlings of *Arabidopsis thaliana* were illuminated for 0 hr (T0), 4 hr (T4), 8 hr (T8), 12 hr (T12), 24 hr (T24), 48 hr (T48), 72 hr (T72), and 96 hr (T96) under white light (40 μmol/m$^2$/s). Hierarchical clustering (Euclidean, average linkage) of normalized protein abundance for photosynthesis-(**A**), galactolipid metabolism- (**B**), chlorophyll metabolism- (**C**),

*Figure 5 continued on next page*

*Figure 5 continued*

and protein import-related proteins during de-etiolation (D). Protein abundance was quantified by shot-gun proteomics and heatmap colors indicate the fold change (average of 3–4 replicates) of each selected protein at each time point of de-etiolation (T0 to T96), relative to the last time point (T96). Note that some PORA values in panel D were higher than 3.5 and outside of the color range limits. Further hierarchical clustering based on the accumulation dynamics of all plastid-localized proteins is provided in *Figure 5—figure supplement 1*.

The online version of this article includes the following source data and figure supplement(s) for figure 5:

**Source data 1.** Chloroplast localized proteins identified by MS and clusters.
**Figure supplement 1.** Accumulation dynamics of selected plastid proteins during de-etiolation.
**Figure supplement 1—source data 1.** List of proteins identified by MS and quantitative data.

ELONGATED HYPOCOTYL 5 (HY5) is a positive regulator of photomorphogenesis, and accumulates during light exposure (*Osterlund and Deng, 1998*). The increase of HY5 peptide abundance was not significant by proteomics but we observed a transient accumulation of the protein between 4 and 72 hr by immunoblot (*Figure 5—source data 1*; *Figure 6*) consistent with the previously reported regulation of abundance during seedling development (*Hardtke et al., 2000*).

We further compared data obtained by proteomics and immunoblot focusing on chloroplast localized proteins. Overall, immunoblot and proteomics provided similar results (*Figure 6* and *Figure 6—figure supplement 1*). PsbA and PsbD (PSII reaction center core), PsbO (Oxygen Evolving Complex), and Lhcb2 (outer antenna complex) proteins were detectable in seedlings at T4, gradually increasing thereafter. Accumulation of the PSI proteins PsaC and PsaD and the Cyt $b_6f$ complex protein PetC started later; these proteins were detectable starting at T8 (*Figure 6A* and *Figure 6—figure supplement 1*). Interestingly, AtpC (ATP synthase complex) was detectable in the etioplast, as described previously (*Plöscher et al., 2011*). Other proteins were selected as markers of etioplast–chloroplast transition. As expected, ELIPs (Early Light Induced Protein) transiently accumulated upon the dark-to-light transition (*Figure 6A*; *Kimura et al., 2003*). As in the proteome analysis, PORA accumulated in etiolated seedlings (T0) and then progressively disappeared upon light exposure. We performed absolute quantification for PsbA, PsaC, and PetC proteins using recombinant proteins as standards (*Figure 6B and C* and *Figure 6—figure supplement 1*). Quantitative data (nmol/seedling) were obtained and normalized using the last time point (*Figure 6C*) to compare the dynamics of protein accumulation. In addition, the comparison of PsbA and PsaC (representative proteins of PSII and PSI, respectively) showed that PsbA levels were about twice that of PsaC at T96 (*Figure 6B and C*).

## Dynamics of chloroplast membrane lipids

Total lipids were extracted from seedlings collected at different time points during de-etiolation (T0, T4, T8, T12, T24, T48, T72, and T96), analyzed by ultra-high-pressure liquid chromatography–mass spectrometry (UHPLC-MS), and quantified against pure standards (*Figure 7—source data 1*). We analyzed the quantity and kinetics of accumulation of 12 different species of galactolipids (*Figure 7A and B*). MGDG 18:3/16:3, MGDG 18:3/18:3, MGDG 18:3/16:1, DGDG 18:3/18:3, and DGDG 18:3/16:0 were the most abundant lipids detected at all time points. Accumulation of all galactolipids increased upon de-etiolation; however, clustering analysis identified two distinct kinetic patterns. One group displayed a leap between T8 and T12, whereas the other group increased later during the de-etiolation period (*Figure 7C*). Interestingly, the two clusters separated the lipids according to the two pathways described for galactolipid synthesis, namely the ER and PL pathways (*Figure 7A and B*; *Marechal et al., 1997*; *Ohlrogge and Browse, 1995*). During early stages of de-etiolation (T0–T24), we observed an incremental accumulation of MGDG and DGDG galactolipids derived from the ER pathway, whereas galactolipids from the PL pathway started to accumulate at T24 (*Figure 7A and B*). The MGDG/DGDG ratio decreased between T0 and T8. This was associated with the transition from PLB (cubic lipid phase) to thylakoid membrane (lamellar structure) (*Bottier et al., 2007*). The MGDG/DGDG ratio started to increase gradually at T8 and was constant by T72 and T96 (*Figure 7D*).

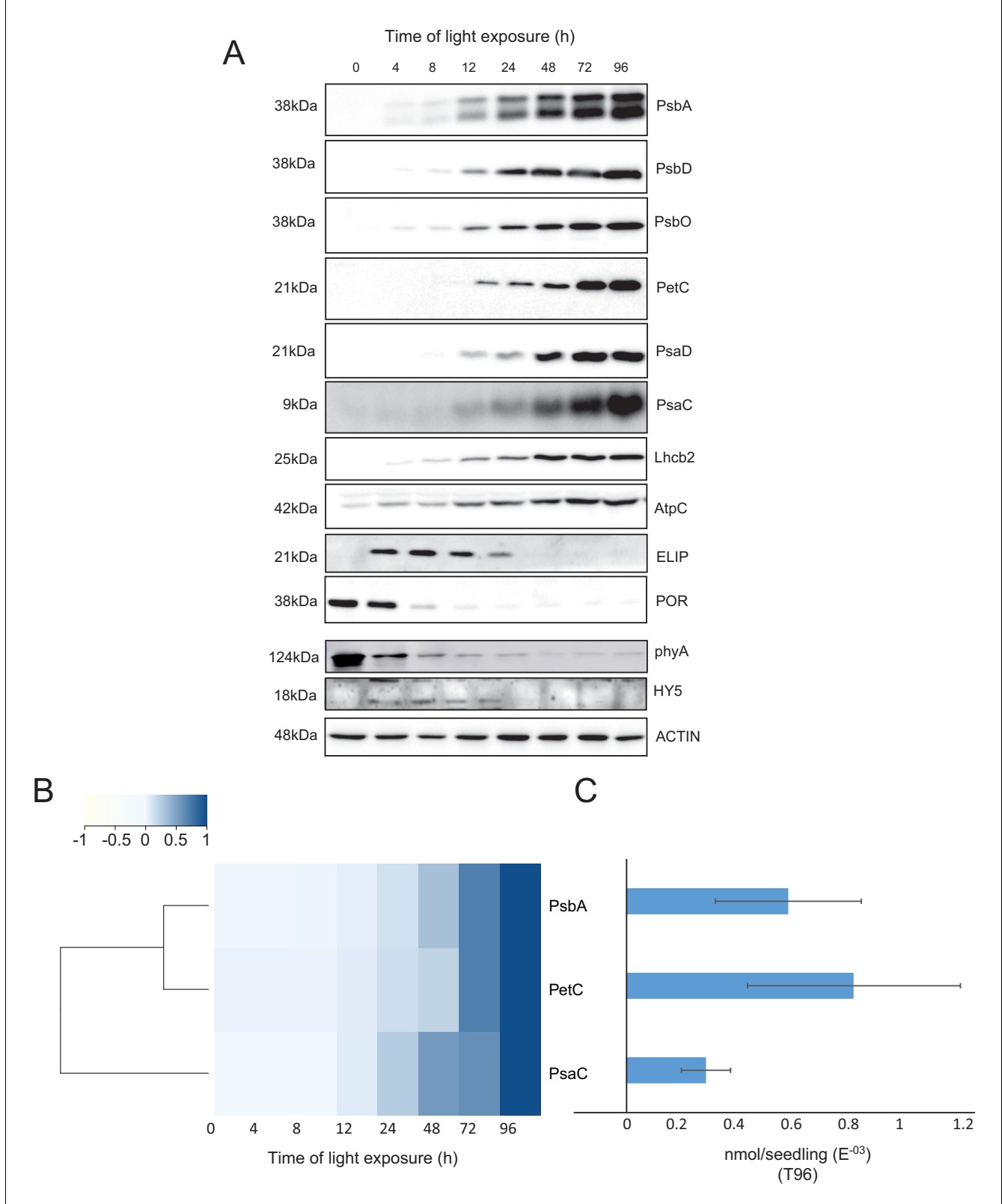

**Figure 6.** Accumulation dynamics of photosynthesis-related proteins during de-etiolation. Three-day-old etiolated seedlings of *Arabidopsis thaliana* were illuminated for 0 hr (T0), 4 hr (T4), 8 hr (T8), 12 hr (T12), 24 hr (T24), 48 hr (T48), 72 hr (T72), and 96 hr (T96) under white light (40 µmol/m²/s). (**A**) Proteins were separated by SDS-PAGE and transferred onto nitrocellulose membrane and immunodetected with antibodies against PsbA, PsbD, PsbO, PetC, PsaD, PsaC, Lhcb2, AtpC, ELIP, POR, phyA, HY5, and ACTIN proteins. (**B–C**) Quantification of PsbA, PetC, and PsaC during de-etiolation.
*Figure 6 continued on next page*

*Figure 6 continued*

Heatmap (**B**) was generated after normalization of the amount of each protein relative to the last time point (T96). Graph (**C**) corresponds to the absolute quantification of proteins at T96. Error bars indicate ± SD (n = 3). Quantification of photosystem-related proteins during de-etiolation is detailed in *Figure 6—figure supplement 1*.

The online version of this article includes the following source data and figure supplement(s) for figure 6:

**Source data 1.** Quantitative data for immunoblot analysis.
**Figure supplement 1.** Quantification of photosynthesis-related proteins.

## Identification of a chloroplast division phase

We observed a massive increase in the accumulation of photosynthesis-related proteins and galactolipids between T24 and T96, corresponding to FC > 2 in the levels of all major chloroplast proteins and lipids (*Figures 6* and *7*). Intriguingly, the total thylakoid surface per chloroplast increased by only 41% between these two time points (*Figure 4A* and *Table 1*). We reasoned that the increase in chloroplast proteins and lipids between T24 and T96 could be explained by increased chloroplast number (per cell and thus per seedling) and thus total thylakoid surface per seedling. We therefore determined chloroplast number per cell and the cell number and volume for each developmental stage through SBF-SEM analysis (T0, T4, T24, and T96) and confocal microscopy analysis for intermediary time points (T24–T96) (*Figure 8* and *Figure 8—figure supplement 1*). The chloroplast number per cell was constant from T4 (25 ± 8) to T24 (26 ± 6); however, in parallel with cell expansion (*Figure 8A and B*), chloroplast number increased sharply (fourfold increase) between T24 (26 ± 6) and T96 (112 ± 29), indicating that two rounds of chloroplast division occurred during this time. Immunoblot analysis of FILAMENTOUS TEMPERATURE-SENSITIVE FtsZ1, FtsZ2-1, and FtsZ2-2 proteins showed that these key components of the chloroplast division machinery were already present during the early time points of de-etiolation. We observed considerably increased accumulation of these proteins between T24 and T48, consistent with the idea that activation of chloroplast division takes place at T24, leading the proliferation of chloroplasts (*Figure 8C*). However, levels of ACCUMULATION AND REPLICATION OF CHLOROPLAST 5 (ARC5) protein, another key component of the chloroplast division machinery, clearly increased during de-etiolation between T8 and T12, presumably reflecting assembly of the chloroplast division machinery before its activation and the proliferation of chloroplasts (*Figure 8D*). To test whether there is a correlation between chloroplast division and either volume or developmental stage, we measured the volume of dividing chloroplasts (selected visually based on the presence of a constriction ring, see *Figure 8—figure supplement 1*) at T24 and T96 using images acquired by SBF-SEM. The average volume of dividing chloroplasts at T24 and T96 were consistently higher than the average volume of all chloroplasts (96 $\mu m^3$ and 136 $\mu m^3$ compared to 62 $\mu m^3$ and 112 $\mu m^3$, respectively) (*Figure 4B*, *Figure 8E* and *Figure 8—source data 1*) indicating that smaller chloroplasts are not dividing. This indicates that developing chloroplasts only divide once a certain chloroplast volume is reached.

## Model of thylakoid surface expansion over time

The quantitative molecular data for the major compounds of thylakoids (galactolipids and proteins) and estimation of chloroplast number per cell allowed us to mathematically determine the thylakoid membrane surface area per seedling and its expansion over time (molecular approach hereafter) and compare it to the surface estimated from the 3D reconstruction (morphometric approach hereafter). First, we calculated the surface area occupied by the main galactolipids (MGDG and DGDG) and photosynthesis-related complexes (PSII, Cyt $b_6f$, and PSI) per seedling (*Table 2*), assuming a 1:1 ratio between number of PsbA, PetC, and PsaC subunits with their corresponding complexes (*Amunts and Nelson, 2009*; *Caffarri et al., 2014*; *Schöttler et al., 2015*).

$$Surface/seedling = nmol/seedling * N * nm^2 \ per \ molecule \qquad (1)$$

Quantitative data for MGDG, DGDG, PsbA, PetC, and PsaC (nmol/seedling) obtained from lipidomic and immunological analyses (*Figures 6* and *7*) were converted into number of molecules/seedling using the Avogadro constant (*N*). To calculate the surface area of outer membrane of thylakoids (i.e. surface exposed to the stroma in lamellae and facing the other thylakoid in appressed regions) and account for the lipid double layer of the membrane, corresponding values of lipids

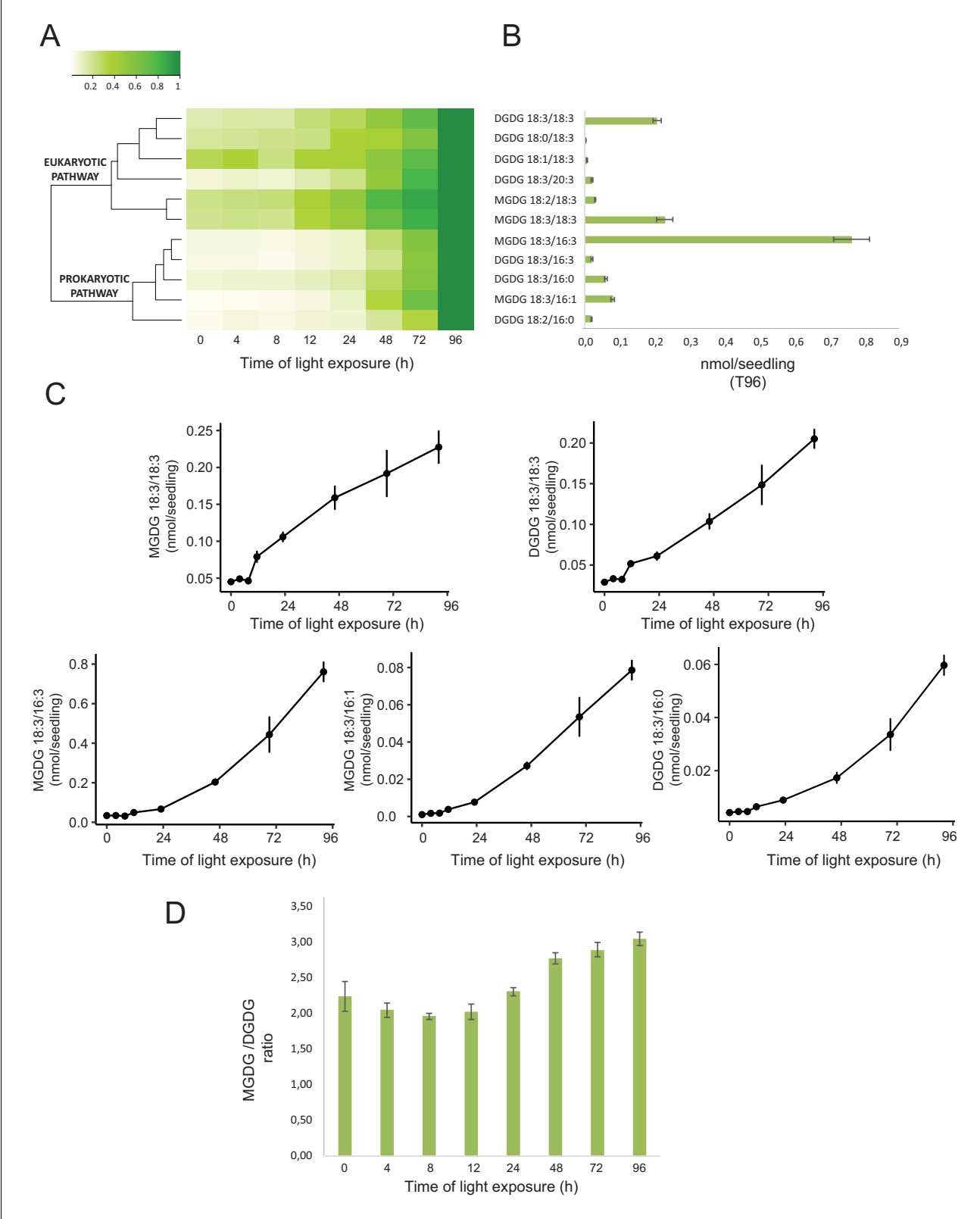

**Figure 7.** Accumulation dynamics of galactolipids during de-etiolation. Three-day-old etiolated seedlings of *Arabidopsis thaliana* were illuminated for 0 hr (T0), 4 hr (T4), 8 hr (T8), 12 hr (T12), 24 hr (T24), 48 hr (T48), 72 hr (T72), and 96 hr (T96) under white light (40 µmol/m²/s). (**A**) Heatmap representation of galactolipids (MGDG and DGDG) during de-etiolation. Samples were normalized to the last time point (T96). (**B**) Absolute quantification at T96 expressed in nmol/seedling. Error bars indicate ± SD (n = 4). (**C**) Absolute quantification (nmol/seedling) of the most abundant chloroplast galactolipids

*Figure 7 continued on next page*

*Figure 7 continued*

MGDG (MGDG 18:3/18:3, MGDG 18:3/16:3, MGDG 18:3/16:1) and DGDG (DGDG 18:3/18:3, DGDG 18:3/16:0) at different time points during de-etiolation. Error bars indicate ± SD (n = 4). (D) The MGDG/DGDG ratio was calculated using all 12 species of galactolipids detected during de-etiolation. Error bars indicate ± SD (n = 4).

The online version of this article includes the following source data for figure 7:

**Source data 1.** Quantitative data for lipidomics.

---

(*Figure 7—source data 1*, *Table 2*) were divided by 2. In addition, the lipid values were corrected by subtracting the portion of lipids incorporated into the envelope rather than present in the thylakoids (*Table 1*). The surface area occupied by molecules of MGDG and DGDG, and that of PSII, Cyt $b_6f$, and PSI photosynthetic complexes ($nm^2$ per molecule, corresponding to stroma-exposed surface) were retrieved from the literature (*Table 3*). Specifically, we used the minimal molecular area of MGDG and DGDG (*Bottier et al., 2007*). To quantify the surface area occupied by the galactolipids and photosynthetic complexes in thylakoids per seedling, the number of molecules per seedling of galactolipids was multiplied by the corresponding molecular surface area, whereas the number of molecules per seedling of PsbA, PetC, and PsaC (subunits of PSII, Cyt $b_6f$, and PSI, respectively) were multiplied by the surface area of the corresponding complex (see *Table 3*).

We calculated thylakoid surface (S) per seedling for each time point (t) as the sum of the surface occupied by MGDG, DGDG, photosynthetic complexes (PS), and ε per seedling, the latter of which corresponds to compounds such as other lipids (e.g. sulfoquinovosyldiacylglycerol, plastoquinone) or protein complexes (ATP synthase and NDH) that were not quantified.

$$S\_thylakoid(t)/seedling = \ (S\_MGDG(t) + S\_DGDG(t) + S\_PS\ (t) + \ \varepsilon \ )/seedling \tag{2}$$

Omitting the unknown ε factor, we plotted the thylakoid surface calculated for each time point where quantitative molecular data were available (T0, T4, T8, T12, T24, T48, T72, and T96) as a function of the duration of light exposure (*Figure 9—figure supplement 1*). The best fitting curve corresponded to a S-shaped logistic function, characterized by a lag phase at early time points (T0–T8), followed by a phase of near-linear increase, and a final plateau at the final time points (T72–T96). To model this function, a four-parameter logistic non-linear regression equation was used to describe the dynamics of the total thylakoid surface over time (*Figure 9—figure supplement 1C*).

## Superimposition of molecular and morphometric data

We compared the values of thylakoid surface, as obtained with the model based on molecular data, with the values obtained from the morphometric analysis (*Figure 9*). The total thylakoid surface per seedling (S_thylakoid_morpho) was calculated by multiplying the thylakoid surface (S_thylakoid) per chloroplast obtained by morphometrics (*Figure 4A*) by the number of chloroplasts (nb.cp) per cell (*Figure 8A*) and the number of cells (nb.cells) per seedlings for each time point (t).

$$\frac{S_{thylakoid_{morpho(t)}}}{seedling} = \tag{3}$$

$$S\_thylakoid(t)/chloroplast \ * \ nb.cp(t)/cell \ * \ nb.cells(t)/seedling$$

We estimated cell number per seedling by measuring the total volume occupied by palisade and spongy cells in cotyledons (that corresponded to 50% of total cotyledon volume; *Figure 9—figure supplement 2*) and dividing this by the average cell volume (*Table 1*). As reported previously (*Pyke and Leech, 1994*), cell number was constant during cotyledon development. We estimated this number as 3000 mesophyll and palisade cells per seedling at T24 and T96 (*Figure 9—figure supplement 2*). The thylakoid membrane surface quantified by the morphometric approach was also estimated at T4, assuming that cell number per cotyledon remained similar between T4 and T24. We compared the thylakoid surface predicted by our mathematical model to the surface estimated experimentally with our 3D thylakoid reconstruction and morphometric measurements (*Figure 9* and *Table 1*). As shown in *Figure 9*, the two approaches showed very similar total thylakoid surface area per seedling at T4 and T24 and differences in this parameter by T96. This indicates that the plateau

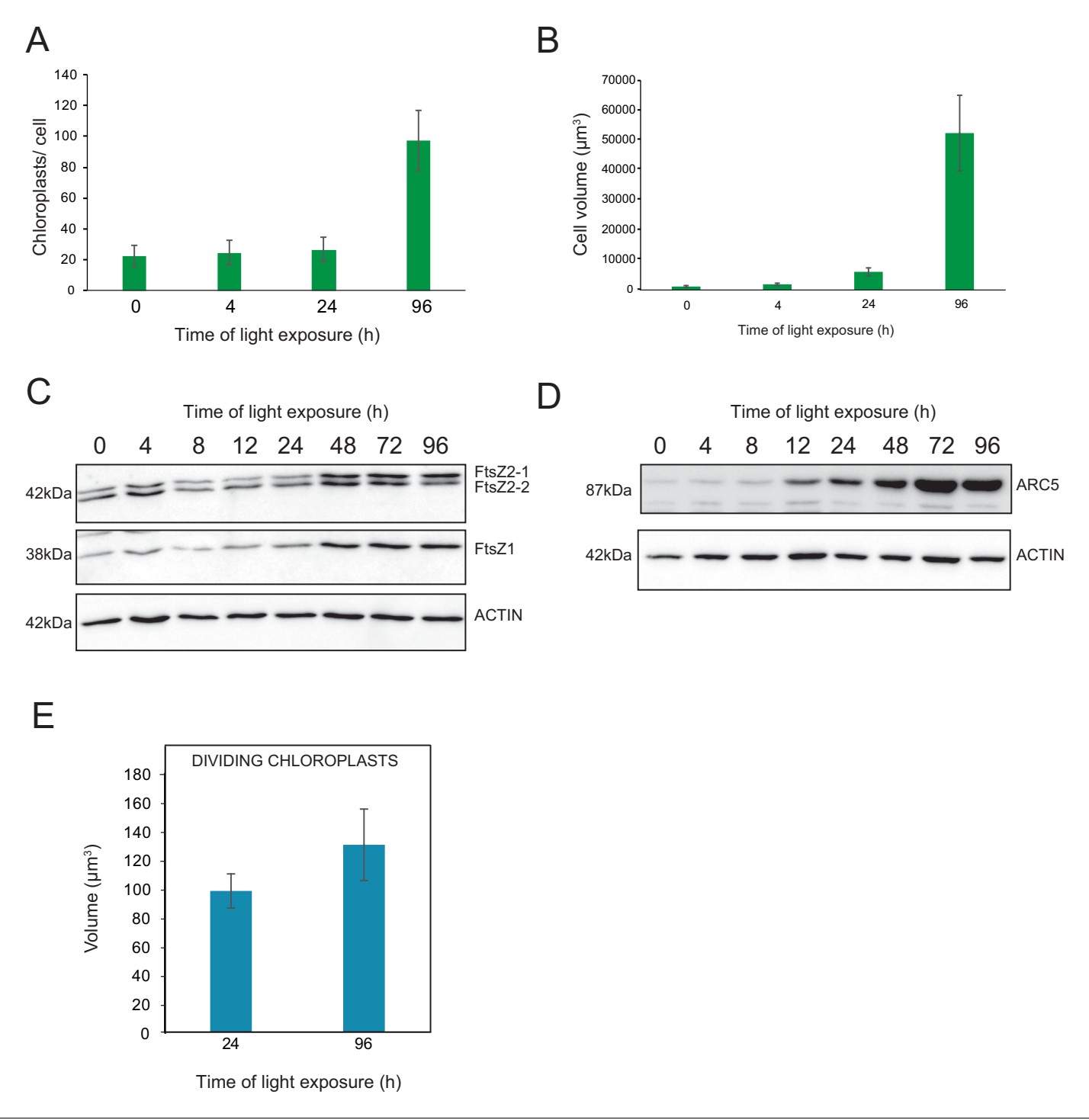

**Figure 8.** Relationship between chloroplast proliferation and chloroplast volume. (A–B) Chloroplast number and cell volume in cotyledons of 3-day-old, dark-grown *Arabidopsis thaliana* seedlings illuminated for 0 hr (T0), 4 hr (T4), 24 hr (T24), and 96 hr (T96) in continuous white light (40 μmol/m²/s). (A) Chloroplast number per cell during de-etiolation. Error bars indicate ± SD (n = 6 for T0 and T4; seven for T24; five for T96). (B) Cell volume was quantified by the Labels analysis module of Amira software. Error bars indicate ± SD (n = 5–6). (C–D) Total proteins were extracted from T0–T96 seedlings, separated on SDS-PAGE, and transferred onto nitrocellulose. Proteins involved in plastid division (C, FtsZ; D, ARC5) and loading control (actin) were detected using specific antibodies (FtsZ2 antibody recognizes both FtsZ2-1 and FtsZ2-2). (E) Volume of dividing chloroplast at T24 and T96. Error bars indicate ± SD (n = 3). Further details of chloroplast proliferation in parallel with cell expansion are provided in *Figure 8—figure supplement 1*.

*Figure 8 continued on next page*

*Figure 8 continued*

The online version of this article includes the following source data and figure supplement(s) for figure 8:

**Source data 1.** Quantitative data for chloroplast number, cell and chloroplast volumes.

**Figure supplement 1.** Chloroplast proliferation in parallel with cell expansion.

phase suggested by the model is not validated and that other components that were not included in the model probably contributed to the expansion of thylakoids at later time points of de-etiolation.

## Discussion

Here, the analysis of 3D structures of entire chloroplasts in Arabidopsis in combination with proteomic and lipidomic analyses provide an overview of thylakoid biogenesis. *Figure 10* depicts a summary of the changes that occur during the de-etiolation process. When considering chloroplast development, our study shows that de-etiolation is divided into two phases. We documented structural changes (disassembly of the PLB and the gradual formation of thylakoid lamellae) and initial increases of ER- and PL-pathway galactolipids and photosynthesis-related proteins (PSII, PSI, and Cyt $b_6f$) during the 'Structure Establishment Phase', which was followed by increased chloroplast number in parallel with cell expansion in the 'Chloroplast Proliferation Phase'. Collection of quantitative data allowed us to create a mathematical model of thylakoid membrane expansion and describe this process during de-etiolation.

### A set of 3D reconstructions of whole chloroplasts by SBF-SEM

In contrast to electron tomography, which is limited in the volume of observation, SBF-SEM allows the acquisition of ultrastructural data from large volumes of mesophyll tissue and the generation of 3D reconstructions of entire cells and chloroplasts (*Figure 3* and *Figure 8—figure supplement 1*, *Videos 1–4*). SEM image resolution was sufficient to visualize stromal lamellae and grana contours, whereas grana segmentation in different lamellae was deduced according to our own TEM analysis and literature data (*Figure 2—figure supplement 1* and *Figure 3—figure supplement 1*). This approach allowed us to obtain quantitative data of chloroplast and thylakoid structure at different developmental stages during de-etiolation at the whole-chloroplast level. By T96, the latest time point of our analysis, the total surface area of thylakoids present in the seedling cotyledons was about 700 mm² (see values in *Table 1* for calculation), about 500-fold greater than the surface area of one cotyledon at this developmental stage. This result is supported by previous estimates made regarding thylakoid surface area relative to leaf surface area (*Bastien et al., 2016*; *Demé et al., 2014*). Moreover, the extent of thylakoid surface area emphasizes how fast and efficient thylakoid

**Table 2.** Surface area occupied by the main galactolipids (MGDG and DGDG) and photosynthetic complexes (PSII, cyt $b_6f$, and PSI). Shown are values at different time points following illumination of 3-day-old etiolated seedlings. Each value indicates the calculated surface area in µm² and corresponds to the average of three biological replicates. Errors indicate SD.

| | T0 | T4 | T8 | T12 | T24 | T48 | T72 | T96 |
|---|---|---|---|---|---|---|---|---|
| MGDG | 1.11E+07 (±0.03E+07) | 1.15E+07 (±0.1E+07) | 1.11E+07 (±0.1E+07) | 1.75E+07 (±0.18E+07) | 4.16E+07 (±0.4E+07) | 8.65E+07 (±0.6E+07) | 1.68E+08 (±0.09E+08) | 2.35E+08 (±0.2E+07) |
| DGDG | 3.64E+06 (±0.4E+06) | 4.23E+06 (±0.5E+06) | 4.10E+06 (±0.1E+06) | 6.26E+06 (±0.5E+05) | 1.32E+07 (±0.107) | 2.32E+07 (±0.2 E+07) | 3.97E+07 (±0.3E+07) | 5.48E+07 (±0.41E+07) |
| PSII | 2.04E+06 (±0.5 E+05) | 2.74E+06 (±0.5E+05) | 4.40E+06 (±0.2E+06) | 9.91E+06 (±1.3E+06) | 2.75E+07 (±0.6E+07) | 6.06E+07 (±0.2E+07) | 1.15E+08 (±0.2E+08) | 1.83E+08 (±0.5E+08) |
| PSI | 0E+00 (±0E+00) | 0E+00 (±0E+00) | 0E+00 (±0E+00) | 8.95E+05 (±4.49E+05) | 1.33E+07 (±0.4E+07) | 2.10E+07 (±1.30E+07) | 3.04E+07 (±0.8E+07) | 4.24E+07 (±1.89E+07) |
| Cyt b6f | 7.99E+05 (±2.33E+05) | 8.43E+05 (±2.91E+05) | 7.5E+05 (±1.33E+05) | 1.57E+06 (±0.7E+06) | 3.44E+06 (±1.22E+06) | 5.30E+06 (±1.01E+06) | 1.69E+07 (±0.5E+06) | 2.37E+07 (±1.11E+07) |

The online version of this article includes the following source data for Table 2:

**Source data 1.** Quantitative data of surface occupied by galactolipids and proteins.

**Table 3.** Surface area occupied by galactolipid and photosynthetic complexes.

Values were retrieved from the corresponding references. MGDG and DGDG surfaces correspond to the minimal molecular area. The surfaces of PSII-LHCII, PSI, and Cyt $b_6f$ complexes correspond to the surface exposed to the stroma (19*26 nm, 20*15 nm, and 90*55 Å, respectively).

|  | Surface in nm2 | reference |
|---|---|---|
| MGDG | 0.82 | *Bottier et al., 2007* |
| DGDG | 0.64 | *Bottier et al., 2007* |
| PSII - LHCII (C2 S2 M2) | 494 | *Caffarri et al., 2014* |
| Cyt b6*f* | 49.5 | *Kurisu et al., 2003* |
| PSI | 300 | *Caffarri et al., 2014* |

biogenesis is during plant development, allowing plants to optimize light absorption capacity, ensuring their primary source of energy.

## Chloroplast development: 'structure establishment phase'

We observed TEM images and quantified 3D chloroplast ultrastructure by SBF-SEM analysis during chloroplast differentiation. Typical etioplast structure of the PLB connected with tubular PTs was replaced by lamellar thylakoids by T4. Measurements of PLB diameter and thylakoid length and thickness were comparable with literature values (*Biswal et al., 2013*; *Daum et al., 2010*; *Kirchhoff et al., 2011*), indicating that these morphometric values are conserved between various model organisms. Thylakoid surface area per chloroplast increased 20-fold between T4 and T24. Remarkably, PSII maximum quantum yield (Fv/Fm) reached the maximal value (0.8) by T14, independent of light intensity (*Figure 1D* and *Figure 1—figure supplement 1*). This shows that PSII

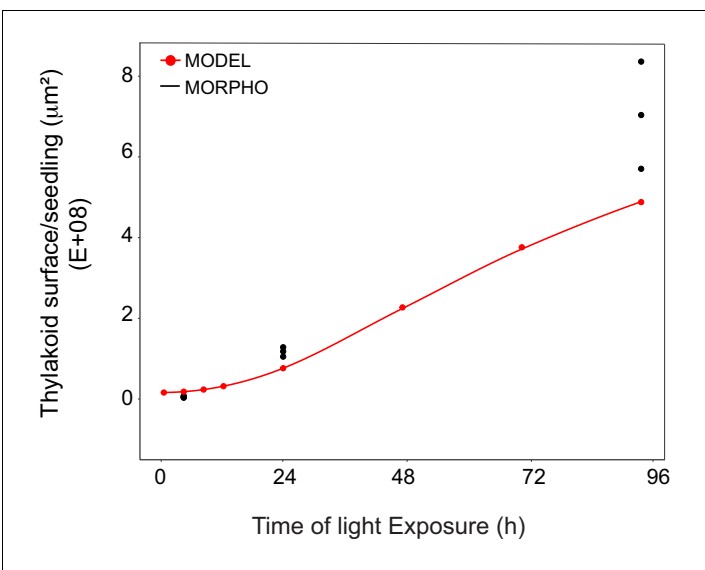

**Figure 9.** Superimposition of thylakoid surface per seedling obtained from morphometric analysis and mathematical modeling. Thylakoid surface per seedling was estimated using quantitative data from 3View analysis ('MORPHO' black dots at T4, T24, and T96; and see *Figure 4* and *Table 1*) and model generated using the quantitative data from proteomics and lipidomics ('MODEL' red line at T0, T4, T8, T12, T24, T48, T72, and T96, and *Table 1*). Further details are provided in *Figure 9—figure supplements 1* and *2*.

The online version of this article includes the following source data and figure supplement(s) for figure 9:

**Source data 1.** Quantitative data used for the mathematical model.
**Figure supplement 1.** Non-linear mixed effect model of thylakoid surface during de-etiolation.
**Figure supplement 2.** Morphometric analysis of cotyledons.

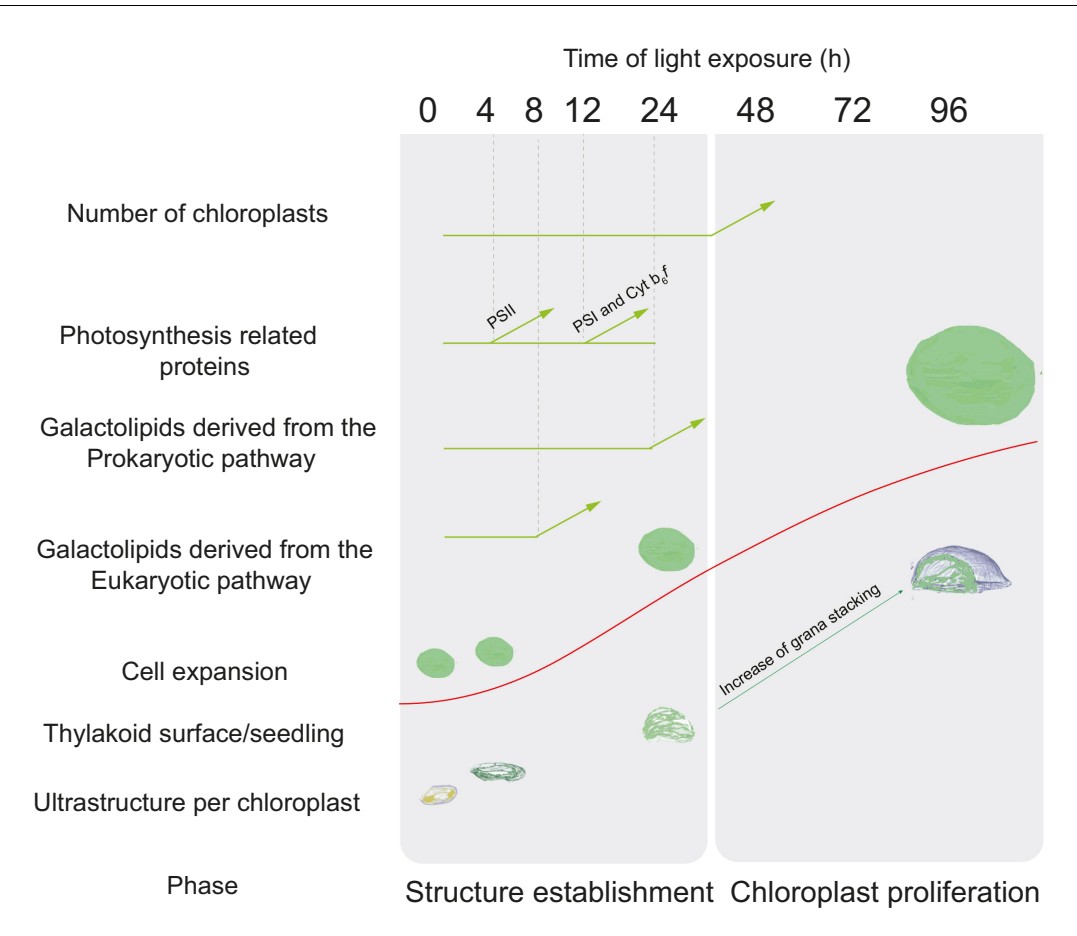

**Figure 10.** Overview of changes observed during the de-etiolation process in *Arabidopsis thaliana* seedlings. The 'Structure Establishment Phase' is correlated with disassembly of the PLB and gradual formation of the thylakoid membrane as well as an initial increase of eukaryotic (after 8 hr) and prokaryotic (after 24 hr) galactolipids and photosynthesis-related proteins (PSII subunits at 4 hr, PSI and cyt $b_6f$ at 12 hr). The subsequent 'Chloroplast Proliferation Phase' is associated with an increase in chloroplast number in concomitance with cell expansion, a linear increase of prokaryotic and eukaryotic galactolipids and photosynthesis-related proteins, and increased grana stacking. The red curve (retrieved from the *Figure 9*) shows thylakoid surface/seedling dynamics during the de-etiolation process.

assembly, and more globally assembly of the photosynthetic machinery, occurs simultaneously with thylakoid membrane formation and that photosynthesis is operational almost immediately upon greening.

Our proteomic and lipidomic analyses suggest that chloroplast ultrastructural changes rely on specifically timed molecular changes. Proteomic analysis revealed the accumulation patterns of more than 5000 unique proteins at eight time points during de-etiolation. These data provide information for plastid development and more widely on light-regulated developmental processes (*Figure 5— source data 1*). Our dataset is more exhaustive regarding temporal resolution and the number of unique proteins detected than that of previous reports on chloroplast differentiation and de-etiolation (*Bräutigam and Weber, 2009*; *Plöscher et al., 2011*; *Reiland et al., 2011*; *Wang et al., 2006*). Overall, the dynamics of the accumulation of proteins revealed by proteomics was similar to the dynamics observed by immunoblots (*Figure 6*; *Figure 6—figure supplement 1* and *Figure 5— source data 1*), although not totally identical for some proteins (e.g. phyA, HY5). The observed differences may be due to the detection methods of the two approaches (detection and relative quantification of individual peptides in proteomics versus detection of the full-length protein by immunoblot analysis) or other inherent limitations of proteomics when faced with low-abundance proteins like transcription factors.

Here, we focused on chloroplast-localized proteins, specifically on thylakoid membrane proteins. According to the SUBA4 localization consensus, 1112 proteins were assigned to plastids, which covers about a third of the total plastid proteome (*Ferro et al., 2003*; *Hooper et al., 2017*; *Kleffmann et al., 2007*). We observed striking changes at the chloroplast ultrastructural levels, and in particular the formation of thylakoids between T0 and T4. However, our proteomic analysis indicated only a few changes in abundance of proteins between these time points, including proteins constituting the photosynthetic machinery (*Figure 8—figure supplement 1* and *Figure 5—source data 1*). Also, we did not observe a significant increase in the major galactolipids constituting the lipid bilayer (*Figure 7* and *Figure 7—source data 1*). Therefore, our data suggest that the reorganization of pre-existing molecules rather than de novo synthesis is responsible for the major chloroplast ultrastructural changes that occur between T0 and T4. These results are consistent with other studies reporting only minor increases in protein accumulation and translation during initial chloroplast differentiation (*Dubreuil et al., 2018*; *Kleffmann et al., 2007*; *Reiland et al., 2011*). A significant change in the proteome was observed when comparing T24 and T0 but overall this change appeared gradual, indicating that increase of chloroplast associated proteins does not exactly follow the two-step induction of corresponding nuclear encoded transcripts reported previously (*Dubreuil et al., 2018*). At T96 the abundance of 607 proteins (12% of the identified) was increased which confirm the massive reorganization of the proteome following the reorganization of the transcriptome during photomorphogenesis (*Ma et al., 2001*). Proteins whose transcript levels decreases in response to light exposure were also downregulated at the protein levels (e.g. phyA and PORA) (*Figure 6*; *Ma et al., 2001*). GO analysis combined with expression pattern–based hierarchical clustering highlighted that most photosynthesis-related proteins are globally coregulated (*Figure 5—figure supplement 1*, clusters 2 and 6) which correlates as well with the overall increase of their corresponding transcripts upon light exposure (*Ma et al., 2001*). However, targeted immunoblot analysis revealed different accumulation dynamics for specific photosystem subunits: PSI subunits were detected at later time points than PSII subunits, but thereafter PSI subunit accumulation was faster (*Figure 6*). The kinetics of different photosynthetic parameters were consistent with the sequential activation of PSII and PSI, in particular photochemical quenching, which showed increased oxidation of the plastoquinone pool by T14 (*Figure 1—figure supplement 1*). Early accumulation of proteins such as Lhcb5, −6, and PSBS could be a way to quickly induce photoprotective mechanisms such as non-photochemical quenching to prevent PSII photodamage during initial photosynthetic machinery assembly. Differences in PSI and PSII accumulation dynamics and activity have been consistently observed in other chloroplast development experimental systems, including in Arabidopsis cell cultures, during germination and development of Arabidopsis seedlings in the light, and in tobacco leaves upon reillumination after dark adaptation (*Armarego-Marriott et al., 2019*; *Dubreuil et al., 2018*; *Liang et al., 2018*). The molecular mechanisms underlying this differential accumulation are currently unknown; however, it is intriguing to observe that PSII protein abundance is higher at early stages of thylakoid formation when grana have not yet been organized. Preferential localization of the PSI and PSII protein complexes in specific thylakoid membrane domains have been reported (lamellae and grana, respectively) (*Wietrzynski et al., 2020*). Therefore, the timing of PSII/PSI relative abundance do not match with their preferential localization. It is possible that the formation of PSI still needs to be delayed until grana formation and PSII relocalization is initiated, which can prevent spillover between the two photosystems (*Anderson, 1981*).

Chloroplast membranes have a specific composition that differs from that of other cell membranes. Galactolipids constitute the bulk of the thylakoid membranes, but are mostly absent from other membrane systems under growth conditions where phosphorus nutrient is available (*Jouhet et al., 2007*). MGDG and DGDG represent around 80% of the thylakoid membrane lipids. The absolute quantification of 12 types of MGDG and DGDG galactolipids (representing the major forms) revealed specific patterns of accumulation (*Figure 7*). Results showed a gradual accumulation of MGDG and DGDG galactolipids derived from the ER pathway from T8 to T24, whereas galactolipids from the PL pathway started to accumulate after 1 day of light exposure (T24). This illustrates the different galactolipid compositions of etioplasts and chloroplasts: ER-pathway galactolipids are predominant in the etioplast whereas PL-pathway galactolipids are predominant in the chloroplast. As no significant changes in lipid accumulation were observed by T4, it appears likely that the emergence of PTs relies on the existing lipids in the etioplast PLB, as suggested also by *Armarego-Marriott et al., 2019*. At later time points, galactolipids from both the ER and PL pathways

constitute the lipid matrix of the thylakoid membrane. How the two galactolipid biosynthesis pathways are regulated during development and/or upon light treatment remains to be elucidated; however, we hypothesize that the PL pathway gains traction after T24 when photosynthetic capacity is fully established.

## Chloroplast development: 'chloroplast proliferation phase'

Chloroplast development continued between T24 and T96, during which thylakoid membranes acquired grana stacks with more clearly defined organization (*Figure 2*). Thylakoid surface increased by only 41%; however, chloroplasts continued to enlarge at a rate comparable to previous de-etiolation stages (T0–T24). This chloroplast volume expansion may be caused by enlargement of extra-thylakoidal spaces occupied by emerging starch granules. These results suggest that large amounts of lipids and proteins are necessary to build up the thylakoid membrane until T24, whereas increases in lipids and proteins between T24 and T96 enable the expansion of already functional thylakoid membranes in preparation for chloroplast division. Indeed, chloroplast number per cell increased during de-etiolation, a process that depends on the division of pre-existing chloroplasts.

Both chloroplasts and mitochondria divide through the activity of supramolecular complexes that constitute the organelle division machineries (*Yoshida, 2018*). As chloroplast proliferation was observed between T24 and T96, chloroplast division may correlate with developmental stage of the organelle. Components of the chloroplast division machinery (e.g. FtsZ and ARC5) were detectable in etioplasts; however, their protein levels accumulated significantly during de-etiolation as chloroplasts proliferated (*Figure 8C and D*). Interestingly, the capacity to divide appeared to correlate with a minimum chloroplast volume of about 100 $\mu m^3$, even at T24 when most chloroplasts were smaller (*Figure 8E* and *Figure 4B*). On the other hand, plastid division and volume seem not to correlate with light and chloroplast photosynthetic capacity in monocots, as etioplasts can divide and increase in size with leaf cell development in absence of light (*Robertson and Laetsch, 1974*; *Klein and Mullet, 1986*). Whether and how cell, chloroplast size and developmental stage can be sensed to activate the chloroplast division machinery remains poorly understood and requires further study.

## A model of thylakoid expansion

Our mathematical model describing the expansion of thylakoid surface per seedling over time considered the surface area occupied by the membrane lipids MGDG and DGDG and the major photosynthetic complexes PSII, PSI, and Cyt $b_6f$. We omitted some components that contribute to the total thylakoid membrane surface (e.g. the protein complexes ATP synthase and NDH, and the lipid sulfoquinovosyldiacylglycerol; together grouped as 'ε' in *Equation 2*). The predictions made by our model fit the surface estimated by SBF-SEM at T4 and T24, whereas they do not fit that at T96. This means that compounds used to generate the mathematical model appear to contribute most to changes in thylakoid surface during early stages of de-etiolation (the structure establishment phase). By contrast, during the later stages of de-etiolation (the chloroplast proliferation phase), the contribution of other compounds omitted in our model is obviously required to build up thylakoid surface.

Our proteomics data (*Figure 5—figure supplement 1* and Dataset 2) revealed some proteins that increased between T24 and T96, such as the FtsH protease (AT2G30950). FtsH proteases have a critical function during thylakoid biogenesis. In Arabidopsis, they constitute a hetero-hexameric complex of four FtsH subunits, which is integrated in the thylakoid membrane (*Kato and Sakamoto, 2018*). Although the FtsH complex surface area is unknown in Arabidopsis, it can be considered as a potential compound contributing to the thylakoid surface changes missing from our mathematical model. Other proteins, such as those involved in carotenoid biosynthesis (AT3G10230) or fatty acid metabolism (AT1G08640), also increased significantly after T24, implying that they contribute to the 'ε' factor.

A follow-up study would be to test the model under different conditions to investigate how this biological system responds to internal (perturbing hormone concentrations, genetic modification of thylakoid lipid and protein composition) or external (different qualities of light) factors. This could be instrumental in revealing new potential regulatory mechanisms of thylakoid biogenesis and maintenance.

Upon de-etiolation, the development of photosynthetic capacity relies on successful chloroplast biogenesis. At the cellular level, this process is expected to be highly coordinated with the metabolism and development of other organelles. Lipid synthesis involves lipid exchanges between chloroplasts and the endoplasmic reticulum. How lipid trafficking is organized remains poorly understood, but could require membrane contact sites between these two organelles (*Michaud and Jouhet, 2019*). Physical interaction between mitochondria and chloroplasts have been reported previously in diatoms (*Bailleul et al., 2015*; *Flori et al., 2017*). Whether such contact sites occur and are functional in plants is unknown; however, these mechanisms are hypothesized to exist since it is necessary that chloroplasts exchange metabolites with mitochondria and peroxisomes to ensure activation of photorespiration concomitantly with photosynthesis. The study of membrane contact sites is an emerging field in cell biology (*Scorrano et al., 2019*). Future work will focus on analysing the dynamics and functionality of contact sites between chloroplast membranes and other organelles, and investigate the general coordination of plant cell metabolism during de-etiolation. These questions could be further addressed using the SBF-SEM stacks and proteomic resource described here.

## Materials and methods

### Plant material and growth conditions

*Arabidopsis thaliana* seeds (Columbia ecotype) were surface-sterilized with 70% (v/v) ethanol with 0.05% (v/v) Triton X-100, then washed with 100% ethanol. Seeds were sown in spots containing 50 seeds (to facilitate rapid harvest) on agar plates containing $0.5 \times$ Murashige and Skoog salt mixture (Duchefa Biochemie, Haarlem, Netherlands) without sucrose. Following stratification in the dark for 3 days at 4°C, seeds were irradiated with 40 µmol $m^{-2}$ $s^{-1}$ for 2 hr at 21°C and then transferred to the dark (plates were covered with three layers of aluminium foil) for 3 days growth at 21°C. For chlorophyll, protein and lipid analyses, 50 etiolated seedlings per time point and replicate were collected in a dark room using a dim green LED lamp as light source (0 hr of light; T0) and at selected time points (T4, T8, T12, T24, T48, T72, T96) upon continuous white light exposure (40 µmol $m^{-2}$ $s^{-1}$ at 21°C), transferred into 1.5 ml tube, flash-frozen in liquid nitrogen and stored at −80°C until further use. For TEM and SBF-SEM microscopy, seedlings were directly immersed into fixation buffer at the corresponding time point.

### Photosynthetic parameters

Maximum quantum yield of photosystem II ($\Phi_{MAX} = F_V/F_M = (Fm-Fo)/Fm$ where Fm is the maximal fluorescence in dark adapted state, Fo is minimal fluorescence in dark adapted state, Fv is the variable fluorescence (Fm-Fo)), photosystem II quantum yield in the light ($\Phi$PSII), and photochemical quenching (qP) were determined using a Fluorcam (Photon Systems Instruments) with blue-light LEDs (470 nm). Plants were dark adapted for a minimum of 5 min before measurement.

### Chlorophyll concentration

Chlorophylls were extracted in 4 volumes of dimethylformamide (DMF) (v/w) overnight at 4°C. After centrifugation, chlorophylls were measured using a NanoDrop instrument at 647 nm and 664 nm. Chlorophyll contents were calculated according to previously described methods (*Porra et al., 1989*).

### Transmission electron microscopy

Samples were fixed under vacuum (200 mBar) in 0.1 M cacodylate buffer (pH 7.4) containing 2.5% (w/v) glutaraldehyde and 2% (w/v) formaldehyde (fresh from paraformaldehyde) for 4 hr and left in the fixation solution for 16 hr at 4°C. Samples were then incubated in a solution containing 3% (w/v) potassium ferrocyanide and 4 mM calcium chloride in 0.1 M cacodylate buffer combined with an equal volume of 4% (w/v) aqueous osmium tetroxide ($OsO_4$) for 1 hr, on ice. After the first heavy metal incubation, samples were rinsed with dd$H_2O$ and treated with 1% (w/v) thiocarbohydrazide solution for 1 hr at 60°C. Samples were rinsed (dd$H_2O$ for 15 min) before the second exposure to 2% (w/v) $OsO_4$ aqueous solution for 30 min at room temperature. Following this second exposure to osmium, tissues were placed in 1% (w/v) uranyl acetate (aqueous) and left overnight at 4°C. The samples were rinsed with dd$H_2O$ for 15 min, and placed in the lead aspartate solution for 30 min at 60°

C. Samples were dehydrated in a series of aqueous ethanol solutions ranging from 50% (v/v) to 100%, then embedded in Durcupan resin by successive changes of Durcupan resin/acetone mixes, with the last imbibition in 100% Durcupan resin. Polymerization of the resin was conducted for 48 hr at 60°C (*Deerinck et al., 2010*). Ultra-thin sections (70 nm) were cut using an Ultrathin-E microtome (Reichert-Jung) equipped with a diamond knife. The sections were analyzed with a Philips CM-100 electron microscope operating at 60 kV.

## Confocal microscopy

To derive the chloroplast and cell volumes, images of 1–5 μm thick sections of cotyledon cells were acquired with ×10 and ×40 oil immersion objectives using a LEICA TCS SP5 confocal laser scanning microscope. Chlorophyll was excited using a red laser (33%) and spectral detection channel was PMT3.

## SBF-SEM

SBF-SEM was performed on Durcupan resin–embedded cotyledons representing the four de-etiolation time points T0, T4, T24, and T96. Overview of the mesophyll tissue ($\approx$600 images) and zoomed stacks of the chloroplasts ($\approx$300 images) were acquired. Voxel size of T4 zoomed stacks: $3.9 \times 3.9 \times 50$ nm; T24: $4.7 \times 4.7 \times 50$ nm; T96: $5.6 \times 5.6 \times 50$ nm. Voxel size for T0 overview: $9.5 \times 9.5 \times 100$ nm; T4: $19.3 \times 19.3 \times 100$ nm; T24: $40 \times 40 \times 200$ nm; T96: $43.5 \times 43.5 \times 200$ nm.

Acquired datasets were aligned and smoothed respectively, using the plugins MultiStackReg and 3D median filter, provided by the open-source software Fiji.

We performed a stack-reslice from Fiji to generate a new stack by reconstructing the slices at a new pixel depth to obtain isotropic voxel size and improve z-resolution. The segmentation and 3D mesh geometry information of plastid /thylakoid (T0, T4, T24 and T96) were implemented by open-source software 3D Slicer (*Fedorov et al., 2012*) and MeshLab (*Cignoni et al., 2008*) respectively.

## Segmentation, 3D reconstruction, and surface and volume quantification

Segmentation and 3D reconstruction of 3View and confocal images were performed using Amira software (FEI Visualization Sciences Group). Specifically, prolamellar body, thylakoids, and envelope membranes as well as the cells were selected using a semi-automatic tool called Segmentation Editor. From the segmented images, triangulated 3D surfaces were created using Generate Surface package. Quantification of morphometric data (Area 3D and volume 3D) was acquired using Label Analysis package.

## Analysis of grana segmentation

Grana structures acquired from SBF-SEM were selected in Amira. The grana selections were converted in line set view in Amira software using the Generate Contour line package. To complete the grana segmentation, the line set views were imported into the Rhino six software (Robert McNeel and Associates, USA). Every granum was segmented in layers with a specific thickness and distance according to quantitative data collected (*Figure 2—figure supplement 1* and *Figure 3—figure supplement 1*). After segmentation, images were re-imported to the Amira software to quantify perimeter using the Label Analysis package.

## Chloroplast number determination

Chloroplasts per cell were counted manually using Image J software (Wayne Rasband, National Institutes of Health). From the same SBF-SEM stack, five and/or 6 cells were cropped at each time point (T0, T4, T24, and T96) to quantify chloroplast number per cell. From TEM images, chloroplast number/cell was determined at T24 (16 cells), T48 (12 cells), T72 (12 cells), and T96 (17 cells). TEM images were acquired from two independent experiments.

## Liquid chromatography–mass spectrometry analysis and protein quantification

Etiolated seedlings were grown as described above. At each time point, ca. 80 seedlings were pooled, frozen in liquid nitrogen, and stored at −80°C until use. Frozen material was ground with a

mortar and pestle, and 40–80 mg of plant material was used for protein and peptide preparation using the iST kit for plant tissues (PreOmics, Germany). Briefly, each sample was resuspended in 100 µL of the provided 'Lysis' buffer and processed with High Intensity Focused Ultrasound (HIFU) for 1 min by setting the ultrasonic amplitude to 65% to enhance solubilization. For each sample, 100 µg of protein was transferred to the cartridge and digested by adding 50 µL of the provided 'Digest' solution. After 180 min of incubation at 37°C, the digestion was stopped with 100 µL of the provided 'Stop' solution. The solutions in the cartridge were removed by centrifugation at 3,800 *g*, whereas the peptides were retained on the iST filter. Finally, the peptides were washed, eluted, dried, and re-solubilized in 18.7 µL of solvent (3% (v/v) acetonitrile, 0.1% (v/v) formic acid).

Mass spectrometry (MS) analysis was performed on a Q Exactive HF-X mass spectrometer (Thermo Scientific) equipped with a Digital PicoView source (New Objective) and coupled to a M-Class UPLC (Waters). Solvent composition at the two channels was 0.1% (v/v) formic acid for channel A and 0.1% formic acid, 99.9% (v/v) acetonitrile for channel B. For each sample, 2 µL of peptides were loaded on a commercial MZ Symmetry C18 Trap Column (100 Å, 5 µm, 180 µm x 20 mm, Waters) followed by nanoEase MZ C18 HSS T3 Column (100 Å, 1.8 µm, 75 µm x 250 mm, Waters). The peptides were eluted at a flow rate of 300 nL/min by a gradient of 8–27% B in 85 min, 35% B in 5 min, and 80% B in 1 min. Samples were acquired in a randomized order. The mass spectrometer was operated in data-dependent mode (DDA), acquiring a full-scan MS spectra (350–1400 m/z) at a resolution of 120,000 at 200 m/z after accumulation to a target value of 3,000,000, followed by HCD (higher-energy collision dissociation) fragmentation on the 20 most intense signals per cycle. HCD spectra were acquired at a resolution of 15,000 using a normalized collision energy of 25 and a maximum injection time of 22 ms. The automatic gain control (AGC) was set to 100,000 ions. Charge state screening was enabled. Singly, unassigned, and charge states higher than seven were rejected. Only precursors with intensity above 250,000 were selected for MS/MS. Precursor masses previously selected for MS/MS measurement were excluded from further selection for 30 s, and the exclusion window was set at 10 ppm. The samples were acquired using internal lock mass calibration on m/z 371.1012 and 445.1200. The mass spectrometry proteomics data were handled using the local laboratory information management system (LIMS) (*Türker et al., 2010*).

Protein quantification based on precursor signal intensity was performed using ProgenesisQI for Proteomics (v4.0.6403.35451; nonlinear dynamics, Waters). Raw MS files were loaded into ProgenesisQI and converted to mzln files. To select the alignment reference, a group of samples that had been measured in the middle of the run (to account for drifts in retention times) and derived from de-etiolation time point T12 or later (to account for increasing sample complexity) was preselected, from which replicate 3 of time point T48 was then automatically chosen as best alignment reference. After automatic peak picking, precursor ions with charges other than 2+, 3+, or 4+ were discarded. The five highest-ranked MS/MS spectra, at most, for each peptide ion were exported, using the deisotoping and charge deconvolution option and limiting the fragment ion count to 200 peaks per MS/MS. The resulting Mascot generic file (.mgf) was searched with Mascot Server version 2.6.2 (http://www.matrixsience.com) using the following settings: trypsin digest with up to two missed cleavages allowed; carbamidomethylation of cysteine as fixed modification; N-terminal acetylation and oxidation of methionine residue as variable modifications; precursor ion mass tolerance 10 ppm; fragment ion (MS/MS) tolerance 0.04 kDa. This search was performed against a forward and reverse (decoy) Araport11 database that included common MS contaminants and iRT peptides. The mascot result was imported into Scaffold Q+S (v4.8.9; Proteome Software Inc), where a spectrum report was created using a false discovery rate (FDR) of 10% and 0.5% at the protein and peptide level, respectively, and a minimum of one identified peptide per protein. After loading the spectrum report into ProgenesisQI, samples were normalized using the 'normalize to all proteins' default settings (i.e. normalization was performed to all ions with charges 2+, 3+ or 4+). Samples were grouped according to de-etiolation time point in a between-group analysis with four replicates for each condition, except for time point T0 and T48, where n = 3. For these two time points, one replicate each had been discarded it appeared as an outlier in principal component analysis (PCA) of protein abundances between different runs (*Figure 5—source data 1*). Quantification employed the Hi-N method, measuring the three most abundant peptides for each protein (*Grossmann et al., 2010*). Associated statistics (p-values, PCA etc.) were calculated in ProgenesisQI, except for the q-values, which were calculated from the p-values using the Benjamini-Hochberg (BH) method, with FDR-adjustment to enforce monotonicity. Quantification also used protein grouping, which assigns proteins for which

only shared but no unique peptides were identified to a 'lead' identifier containing all these shared peptides and thus having the greatest coverage among all grouped identifiers or highest score where coverage is equal. Quantification was restricted to protein (groups) with at least two identified peptides among which at least one is unique to the protein (group). Using these requirements, 5082 Arabidopsis proteins (or groups) were identified. Since 13 additional identifications were exclusively associated with decoy proteins, the false discovery rate at the protein level is estimated to be 0.3%. The mass spectrometry proteomics data have been deposited to the ProteomeXchange Consortium via the PRIDE (*Perez-Riverol et al., 2019*) partner repository with the dataset identifier PXD021518.

## Immunoblot analysis

Proteins were extracted from whole seedlings in four volumes (w/v) of SDS-PAGE sample buffer (0.2 M Tris/HCL pH 6.8, 0.4 M dithiothreitol, 8% (w/v) SDS, 0.4% (w/v) Bromophenol blue, and 40% (v/v) glycerol).

Proteins were denatured for 15 min at 65°C and cell debris were removed by centrifugation for 5 min at 16,000 $g$. Proteins were separated on SDS-PAGE (10–15% (w/v) polyacrylamide concentrations depending on the molecular weight of the protein of interest) and transferred onto a nitrocellulose membrane for immunoblotting (overnight at 4°C) in Dunn buffer (10 mM NaHCO$_3$, 3 mM Na$_2$CO$_3$, 0.01% (w/v) SDS, and 20% ethanol).

Absolute quantification of PsbA, PetC, and PsaC was performed according to Agrisera instructions and using recombinant proteins (PsbA AS01 0116S, PetC AS08 330S, and PsaC AS04 042S; Agrisera, Vännäs, SWEDEN). Three respective calibration curves for the three recombinant proteins were created. Concentrations used to generate the PsbA and PetC calibration curves were 1.75, 2.5, 5, and 10 (ng/μL). Concentrations used to generate the PsaC calibration curve were 0.375, 0.75, 1.5, and 3 (ng/μL). Immunodetections were performed using specific antibodies: anti-Actin (Sigma, A0 480) at 1/3000 dilution in 5% (w/v) milk in Tris-buffered saline (TBS); anti-Lhcb2 (Agrisera, AS01 003), anti-D1(PsbA) (Agrisera, AS05 084), anti-PsbO (Agrisera, AS14 2825), anti-PsbD (Agrisera, AS06 146), anti-PetC (Agrisera, AS08 330), and anti-AtpC (Agrisera, AS08 312) at 1/5000 dilution in 5% milk/TBS; Anti-PsaD (Agrisera, AS09 461) at 1/2000 in 5% milk/TBS; and anti-PsaC (Agrisera, AS042P) and anti-ARC5 (Agrisera, AS13 2676) at 1/2000 in 3% (w/v) bovine serum albumin (BSA) in TBS. Anti-FtsZ-1 and anti-FtsZ2-1/FtsZ 2–2 (*El-Shami et al., 2002*; *Karamoko et al., 2011*) were used at 1/2000 dilution in 5% milk/TBS. After incubation with primary antibodies overnight at 4°C, blots were washed three times in TBS containing 0.1% (v/v) Tween without antibodies for 10 min and incubated for 1 hr at RT with horseradish peroxidase–conjugated secondary antibodies (1/3000 (v/v) anti-rabbit or anti-mouse secondary antibodies, Agrisera). For Anti-HY5 (1/1000 dilution; *Oravecz et al., 2006*) and anti-phyA (1/1000 dilution; *Shinomura et al., 1996*), TBS was replaced by Phosphate Buffer Saline (PBS). Chemiluminescence signals were generated with Enhanced chemiluminescence reagent (1 M Tris/HCl pH 8.5, 90 mM coumaric acid, and 250 mM luminol) and detected with a Fujifilm Image – Quant LAS 4000 mini CCD (GE Healthcare). Quantifications were performed with ImageQuant TL software (GE Healthcare).

## Lipid profiling

Lipids were extracted from whole seedlings ground in a mortar and pestle under liquid nitrogen. Ground plant material corresponding to 40–80 mg fresh weight was suspended in tetrahydrofuran: methanol (THF/MeOH) 50:50 (v/v). 10–15 glass beads (1 mm in diameter) were added followed by homogenization (3 min, 30 Hz,) and centrifugation (3 min, 14 000 $g$, at 4°C). The supernatant was removed and transferred to an HPLC vial. Lipid profiling was carried out by ultra-high pressure liquid chromatography coupled with atmospheric pressure chemical ionization-quadrupole time-of-flight mass spectrometry (UHPLC-APCI-QTOF-MS; *Martinis et al., 2011*). Reverse-phase separation was performed at 60°C on an Acquity BEH C18 column (50 × 2.1 mm, 1.7 μm). The conditions were the following: solvent A = water; solvent B = methanol; 80–100% B in 3 min, 100% B for 2 min, re-equilibration at 80% B for 0.5 min. Flow rate was 0.8 ml min$^{-1}$ and the injection volume 2.5 μl. Data were acquired using MassLynx version 4.1 (Waters), and processed with MarkerLynx XS (Waters). Peak lists consisting of variables described by mass-to-charge ratio and retention time were generated (*Martinis et al., 2011*; *Spicher et al., 2016*).

Absolute quantification of mono- (MGDG) and di-galactosyldiacylglycerol (DGDG) was conducted by creating calibration curves using MGDG (reference number 840523) and DGDG (reference number 840523) products of Avanti Company. Calibration curves were prepared using the following concentrations: 0.08, 0.4, 2, 10, and 50 µg ml$^{-1}$ of MGDG or DGDG.

## Mathematical model

A non-linear mixed effects model (with fixed effect of time and random effect of replicates on 3 of the parameters), built on a four-parameter logistic function, was implemented in R (free software created by Ross Ihaka and Robert Gentleman, Auckland University, New Zealand), following the examples in *Pinheiro and Bates, 2000*. The R-packages used are: nlme (*Pinheiro and Bates, 2000*), effects, lattice and car (*Fox and Weisberg, 2018*). To account for self-correlation at the replicate level, we proceeded to fit an overall mixed-effects model to the data (package 'nlme' from R), using the replicate's as random effect term (*Figure 9—figure supplement 1*). The four parameters *a, b, c,* and *d* have been calculated (*Figure 9—figure supplement 1*) and the three plots (one for each biological replicate) (*Figure 9—figure supplement 1*) indicated the fitting curve for a series of data points.

## Acknowledgements

This work was supported by the University of Neuchâtel and ETH Zurich, a grant from the Swiss National Science Foundation (3100A0-112638) to ED, and grants 31003A_156998 and 31003A_176191 to FK. We thank Jonas Grossmann, Laura Kunz and Paolo Nanni from the Functional Genomics Center Zurich (FGCZ) for peptide preparation for mass spectrometry, acquisition of the raw data and help with associated data analysis, the ETH Zurich microscopy facility (ScopeM) for advice in conducting SBF-SEM analysis. We thank Slobodeanu Radu Alexandru and Federico Giacomarra for help with bioinformatics analysis, and Romain Bessire for help with image processing software. We thank Christian Fankhauser for kindly providing anti-phyA antibodies. We thank Roman Ulm and Michel Goldschmidt-Clermont for critical reading of the manuscript.

## Additional information

### Funding

| Funder | Grant reference number | Author |
|---|---|---|
| Schweizerischer Nationalfonds zur Förderung der Wissenschaftlichen Forschung | 3100A0-112638 | Emilie Demarsy |
| Schweizerischer Nationalfonds zur Förderung der Wissenschaftlichen Forschung | 31003A_156998 | Felix Kessler |
| Schweizerischer Nationalfonds zur Förderung der Wissenschaftlichen Forschung | 31003A_176191 | Felix Kessler |

The funders had no role in study design, data collection and interpretation, or the decision to submit the work for publication.

### Author contributions

Rosa Pipitone, Conceptualization, Data curation, Software, Formal analysis, Validation, Investigation, Visualization, Methodology, Writing - original draft, Writing - review and editing; Simona Eicke, Barbara Pfister, Conceptualization, Data curation, Formal analysis, Validation, Investigation, Visualization, Methodology, Writing - review and editing; Gaetan Glauser, Conceptualization, Resources, Formal analysis, Validation, Methodology, Writing - review and editing; Denis Falconet, Resources, Data curation, Formal analysis, Validation, Investigation, Visualization, Methodology, Writing - review and editing; Clarisse Uwizeye, Conceptualization, Data curation, Software, Formal analysis, Investigation, Visualization, Methodology, Writing - review and editing; Thibaut Pralon, Formal analysis, Investigation, Writing - review and editing; Samuel C Zeeman, Conceptualization, Resources, Funding

acquisition, Validation, Writing - review and editing; Felix Kessler, Conceptualization, Resources, Supervision, Funding acquisition, Validation, Project administration, Writing - review and editing; Emilie Demarsy, Conceptualization, Resources, Data curation, Formal analysis, Supervision, Funding acquisition, Validation, Investigation, Visualization, Methodology, Writing - original draft, Project administration, Writing - review and editing

## Author ORCIDs
Rosa Pipitone (ID) https://orcid.org/0000-0002-6855-5614
Simona Eicke (ID) http://orcid.org/0000-0003-4180-2440
Barbara Pfister (ID) https://orcid.org/0000-0002-4183-9625
Gaetan Glauser (ID) https://orcid.org/0000-0002-0983-8614
Denis Falconet (ID) http://orcid.org/0000-0001-8182-1182
Clarisse Uwizeye (ID) https://orcid.org/0000-0002-5323-6098
Thibaut Pralon (ID) https://orcid.org/0000-0001-6029-1565
Samuel C Zeeman (ID) https://orcid.org/0000-0002-2791-0915
Felix Kessler (ID) https://orcid.org/0000-0001-6409-5043
Emilie Demarsy (ID) https://orcid.org/0000-0002-0638-0812

## Decision letter and Author response
Decision letter https://doi.org/10.7554/eLife.62709.sa1
Author response https://doi.org/10.7554/eLife.62709.sa2

## Additional files

### Supplementary files
• Transparent reporting form

### Data availability
All data generated or analysed during this study are included in the manuscript and supporting files. Sources data files for Figures 4, 5 6 7 8 and 9 and associated figure supplements have been provided.

The following dataset was generated:

| Author(s) | Year | Dataset title | Dataset URL | Database and Identifier |
|---|---|---|---|---|
| Pfister B, Zeeman SC, Pipitone R, Kessler F, Demarsy E | 2021 | Proteomics of Arabidopsis seedlings during de-etiolation | https://www.ebi.ac.uk/pride/archive/projects/PXD021518 | PRIDE, PXD021518 |

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
