## [Decision Letter]

**Acceptance summary:**

The reviewers and myself are impressed by the your unprecedented, comprehensive and multifaceted analysis of chloroplast development, which provides a rich resource for plant biologists. The technically challenging 3D characterization of time resolved thylakoid development and its correlation with protein and lipid abundance allows a new level of understanding of chloroplast development and the conceptual division of this process into two phases.

**Decision letter after peer review:**

Thank you for submitting your article "Two distinct phases of chloroplast biogenesis during de-etiolation in *Arabidopsis thaliana*" for consideration by *eLife*. Your article has been reviewed by three peer reviewers, and the evaluation has been overseen by a Reviewing Editor and Christian Hardtke as the Senior Editor. The following individuals involved in review of your submission have agreed to reveal their identity: Thomas Pfannschmidt (Reviewer #1); R. Paul Jarvis (Reviewer #2).

The reviewers have discussed the reviews with one another and the Reviewing Editor has drafted this decision to help you prepare a revised submission.

The editors have judged that your manuscript is of interest, but as described below that few additional experiments and analyses are required before it is published, we would like to draw your attention to changes in our revision policy that we have made in response to COVID-19 (https://elifesciences.org/articles/57162). First, because many researchers have temporarily lost access to the labs, we will give authors as much time as they need to submit revised manuscripts. We are also offering, if you choose, to post the manuscript to bioRxiv (if it is not already there) along with this decision letter and a formal designation that the manuscript is "in revision at *eLife*". Please let us know if you would like to pursue this option. (If your work is more suitable for medRxiv, you will need to post the preprint yourself, as the mechanisms for us to do so are still in development.)

Summary:

All three reviewers as well as myself are impressed by the in depth and multi-method analysis of chloroplast and thylakoid membrane development provided in your study, including time courses of 3D imaging combining TEM, SBF-SEM and confocal microscopy, lipidomics and proteomics. However, some analyses need to be improved and/or better explained.

Essential revisions:

There is a concern about the proteomics analysis, as the low number of proteins changing in abundance upon de-etiolation is unexpected. It is not clear how the samples were harvested. Were they harvested in the light and could that have influenced protein abundance? The harvesting procedure needs to be better explained. Or is the proteomics method not sensitive enough? The proteomics should to be validated, for example by Western Blots with well-established marker proteins such as phyA and HY5.

Please also add loading controls to Figure 6 and the associated supplemental figure.

Please explain better how the volume of dividing chloroplasts was determined.

Reviewer #1:

The work by Pipitone et al. is a very carefully performed and technically sophisticated elucidation of the establishment of the thylakoid membrane system in Arabidopsis chloroplasts upon first illumination of cotyledons. Its charm is the three-dimensional resolution during a time course that allows to follow the rapid changes occurring during the short time window in which the greening occurs. In addition authors included proteomics and lipidomics approaches complementing the morphological observations by sound molecular data. All together the study provides a very detailed catalogue of the processes that trigger chloroplast biogenesis that is highly useful for the community as it provides important numbers of size and development.

Improvements

Actually the work has been performed very carefully and there is not much to improve.

The Introduction could contain more references.

SBF-SEM should be spelled out at first mentioning and maybe a bit more background about the technology would be helpful for the reader to understand it.

Subsection “Quantitative analysis of thylakoid surface area per chloroplast during de-etiolation”: The occurrence of starch granules is of course caused by the continuous illumination. It however may have also an impact on the final size of the plastid. It would be interesting to know whether chloroplasts at the end of a night phase are smaller than at the end of a light phase. This is not mandatory for the current manuscript but an interesting question to follow in future and maybe to be discussed.

“The surface area…” please define what is meant since membranes have two sides.

Subsection “Quantitative analysis of thylakoid surface area per chloroplast during de-etiolation” second paragraph: There is another study done in cell culture that has a similar design (Dubreuil et al. ), are the two studies compatible with each other in their conclusion and if not, what are the differences?

“Our data suggest that the reorganisation of pre-existing molecules rather than de novo synthesis is responsible for the major chloroplast ultrastructural changes that occur between T0 and T4.”: This sentence is not perfectly clear to me. Maybe the authors can explain this a bit more in detail using examples.

Subsection “Chloroplast development: ‘Structure Establishment Phase’”: I think it is worth noting that the interactions between PSII complexes located in neighbouring thylakoid membranes trigger the stacking of the grana. It is therefore tempting to speculate that stroma lamellae are established first and that these membranes are then stacked after PSII complexes are inserted into the membrane because they provide the adhesion points between them.

Reviewer #2:

This impressive manuscript describes a comprehensive, multifaceted analysis of the morphological and molecular changes that accompany photosynthetic establishment during seedling de-etiolation. Morphological data, focusing in particular on the photosynthetic thylakoid membranes, are derived using transmission electron microscopy (TEM), serial block face scanning electron microscopy (SBF-SEM), and confocal microscopy, while quantitative molecular data on the abundancies of proteins and lipids are derived using mass spectrometry and western blotting. The various data are acquired over a time course between 0 h and 96 h post illumination, and with a high level of temporal resolution. The data allow the authors to develop a mathematical model for the expansion of the surface area of thylakoids (reaching 500-times the surface area of the cotyledon leaf), which matches well with experimental observations from the SBF-SEM analysis for earlier, but not later, stages of de-etiolation. Moreover, the data point to a two-phase organization of the de-etiolation process, with the first phase ("Structure Establishment") characterized by thylakoid assembly and photosynthetic establishment, and the second phase ("Chloroplast Proliferation") characterized by chloroplast division and cell expansion.

The data are of a high standard, and the depth and breadth of analysis in a single, unified study is unprecedented. While it is arguable that there are few major, completely novel insights reported here (indeed, in the Discussion, the authors very helpfully point out how many of the parameters they have measured are consistent with data reported elsewhere by others), this should not detract from the overall value of the study; a major and unique strength here is that all of the data have been acquired together and so are directly comparable. I have no doubt that this dataset will be extremely interesting to many researchers, and prove to be an invaluable resource for the plant science community. Consequently, I am sure that it will attract many citations.

I have a few specific comments that I would like the authors to consider carefully, as follows.

1) Figure 3. The 3D reconstructions are undoubtedly useful for deriving quantitative data, as they enabled the derivation of thylakoid surface area data to verify the mathematical model. However, it is very difficult to see anything clearly in the images shown in the figure. I wonder if the authors can make the images clearer, and then also point to and describe some of the key features. The videos do help a bit, but even these are not that clear.

2) Subsection “Quantitative analysis of thylakoid surface area per chloroplast during de-etiolation”, second paragraph. It is here that the "two phases" model is first proposed. I really could not see a clear basis for proposing this model here, using the data that had been presented thus far. As I see it (and based on the way the two phases are described in the Discussion), one can't really propose this model until after the chloroplast number and cell size data have been presented.

Moreover, the description of the second phase here ("and a second phase…") seems a bit inconsistent with the statement in the paragraph above that thylakoid surface area increases dramatically between T4 and T24, and much less between T24 and T96.

3) Figure 6, and the related supplementary figure. Loading controls are missing here, and should be added. Also, it is stated that a number of proteins (PsbA, PsbD, PsbO, Lhcb2) are "detectable" at T0. To me, they look UNdetectable.

4) Dividing chloroplasts. In subsection “Identification of a chloroplast division phase”, it is stated that the volume of dividing chloroplasts was measured, and we are referred to Figures 8E and 4B in support of this statement. However, it is not explained how this was done. More clear and specific explanation is needed. Was it the case that the authors sought out and measured dumbbell-shaped organelles, and quantified those? If so, images are needed to illustrate this point. And, I don't see anything relevant in Figure 4B – this callout apparently belongs in the following sentence. The statement that the average size of dividing chloroplasts was higher than that of all chloroplasts is not really surprising if the authors were measuring organelles just on the point of becoming two organelles.

5) Subsection “Model of thylakoid surface expansion over time”. The motivation for this section needs to be better introduced. When I first read it, I could not understand why the authors wished to again "determine the thylakoid membrane surface area", as this had already been discussed earlier in the manuscript.

Also related to the modelling: Did the authors take into account the existence of appressed membranes when calculating the surface area exposed to the cytosol. And, assuming it is clearly established that there is a 1:1 relationship between these proteins and the relevant complexes, perhaps this should be stated and the relevant literature cited.

Reviewer #3:

This study by Pipitone et al. combines SBF-SEM microscopy with quantitative proteomics and lipidomics to explore chloroplast differentiation. Authors describe that chloroplast biogenesis occurs in a first phase of structure establishment with thylakoid biogenesis, followed by a second phase of chloroplast division. The images and 3D reconstructions are beautiful, the quantitative data are novel, and their integration offers a new perspective into the seedling de-etiolation process, a model system for physiological and molecular studies. However, in my opinion some aspects need to be better explained and significantly improved.

The authors write: "After 8h of illumination (T8), we observed decreased abundance of only one protein (the photoreceptor cryptochrome 2, consistent with its photolabile property) and increased levels of only three proteins, which belonged to the chlorophyll a/b binding proteins category involved in photoprotection (AT1G44575 = PsbS; AT4G10340 = Lhcb5; AT1G15820 = Lhcb6"). This is striking, as many well studied proteins change in abundance during the first hours of de-etiolation. Actually, looking into the data set with the quantification data for of the ~5,000 proteins, it appears that many proteins do show significant changes between T0 and T8. For example PORA and ELIP, changes that are also reflected in Figure 6A.

Related to the above, well known proteins for example phyA and HY5, that undergo drastic changes in abundance when etiolated seedlings are first exposed to light, do not show changes in T4,T8 and T12 relative to T0 in the proteomics data set. This raises questions about the proteomic approach (sensitivity of the method?) or the experimental setup. Could authors please comment on this? I feel that validation of the proteomics approach is critical, especially taking into account the central conclusion that "the first 12h of illumination saw very few significant changes in protein abundance".

Subsection “Chloroplast development: ‘Structure Establishment Phase’” paragraph three: A reference is needed. Also, it is mentioned that PSII appears later than PSI, which does not seem to match the observation that PSII proteins appear earlier than PSI, or that the surface area occupied at early time points by PSII is greater than the one occupied by PSI. Please check.

Are the calculations of thylakoid surface expansion over time consistent with previous available data using tomography? Please include.

In the Introduction, authors could include mention of the massive transcriptional reprogramming that take place during de-etiolation. In addition, I think that comparison of the proteomics data with the transcriptomic changes during de-etiolation (well described in the literature) would allow further understanding of the distinct phases proposed. For the chloroplast proteins already present in the dark, how does this correlate with expression of the corresponding genes?

---

## [Author Response]

Essential revisions:There is a concern about the proteomics analysis, as the low number of proteins changing in abundance upon de-etiolation is unexpected. It is not clear how the samples were harvested. Were they harvested in the light and could that have influenced protein abundance? The harvesting procedure needs to be better explained.

We now added a detailed description of the harvesting procedures in the Material and methods section which now reads: “Seeds were sown in spot containing 50 seeds (to facilitate rapid harvest) on agar plates containing 0.5 × Murashige and Skoog salt mixture (Duchefa Biochemie, Haarlem, Netherlands) without sucrose. Following stratification in the dark for 3 days at 4°C, seeds were irradiated with 40 μmol m-2 s^-1^ for 2 h at 21°C and then transferred to the dark (plates were covered with 3 layers of aluminium foil) for 3 days growth at 21°C. For chlorophyll, protein and lipid analyses, 50 etiolated seedlings per time point and replicate were collected in a dark room using a dim green LED lamp as light source (0 h of light; T0) and at selected time points (T4, T8, T12, T24, T48, T72, T96) upon continuous white light exposure (40 μmol m-2 s^-1^ at 21°C), transferred into 1.5ml tube, flash-frozen in liquid nitrogen and stored at 80°C until further use. For TEM and SBF-SEM microscopy, seedlings were directly immersed into fixation buffer at the corresponding time point.”

Or is the proteomics method not sensitive enough? The proteomics should to be validated, for example by Western Blots with well-established marker proteins such as phyA and HY5.

The proteomics method indicates the abundance of a specific protein based on the detection and quantification of representative peptides. Data related to chloroplast localized proteins were consistent when comparing the results from proteomics method and immunoblotting approach and in agreement with previous studies (subsection “Dynamics of plastid proteins related to thylakoid biogenesis”; Figure 6 and its corresponding figure supplement 1). To further validate our proteomics data, which for low abundance proteins can be less precise than targeted measurements, we now provide western blots for phyA and HY5 (Figure 6—figure supplement 1). We have added the description of the dynamics of phyA and HY5 abundance revealed by western blots and proteomics and discussed the corresponding data.

Please also add loading controls to Figure 6 and the associated supplemental figure.

We used actin as a loading control and added the data to Figure 6.

Quantitative western (Figure 6—figure supplement 1) were performed using 3 to 4 replicates from independent experiments, we did not use loading control but normalization was based on fresh weight (and corresponding number of seedlings). Standard deviation includes potential variations of loadings between experiments. All raw data are now provided in (Figure 6—source data 1).

Please explain better how the volume of dividing chloroplasts was determined.

Chloroplasts in division were selected visually based on the presence of a constriction ring. We now show examples of chloroplasts in division used for this analysis (3D reconstruction of one T24 and two T96 chloroplasts, Figure 8—figure supplement 1 panel P). The 3D volume of selected chloroplast was determined using the label analysis package of Amira software as indicated in Materials and methods.

Reviewer #1:The work by Pipitone et al. is a very carefully performed and technically sophisticated elucidation of the establishment of the thylakoid membrane system in Arabidopsis chloroplasts upon first illumination of cotyledons. Its charm is the three-dimensional resolution during a time course that allows to follow the rapid changes occurring during the short time window in which the greening occurs. In addition authors included proteomics and lipidomics approaches complementing the morphological observations by sound molecular data. All together the study provides a very detailed catalogue of the processes that trigger chloroplast biogenesis that is highly useful for the community as it provides important numbers of size and development.ImprovementsActually the work has been performed very carefully and there is not much to improve.The Introduction could contain more references.

Done as suggested.

SBF-SEM should be spelled out at first mentioning and maybe a bit more background about the technology would be helpful for the reader to understand it.

We now added a description of the technique. The text now reads : “Serial Block Face-Scanning Electron Microscopy (SBF-SEM) is a technique where the embedded specimen is imaged by scanning the face of the block with an electron beam. After imaging, the face of the block is shaved automatically (e.g. 60 nm-thick slices) by an ultramicrotome mounted in the vacuum chamber. The section is discarded and the newly revealed block face is imaged again. Repeated imaging and cutting allows the collection of a tomographic sequence of hundreds of images of the same area. Thereby, a much larger volume can be reconstructed in 3D to show cellular organisation (Peddie and Collinson, 2014; Pinali and Kitmitto, 2014). “.

Subsection “Quantitative analysis of thylakoid surface area per chloroplast during de-etiolation”: The occurrence of starch granules is of course caused by the continuous illumination. It however may have also an impact on the final size of the plastid. It would be interesting to know whether chloroplasts at the end of a night phase are smaller than at the end of a light phase. This is not mandatory for the current manuscript but an interesting question to follow in future and maybe to be discussed.

We agree with the reviewer that the observed enlargement of chloroplast after 24h of illumination might in part be due to increased volume occupied by starch granule, and we integrate this explanation in the text: “This chloroplast volume expansion may be caused by enlargement of extra-thylakoidal spaces occupied by emerging starch granules”, however we did not try to validate this hypothesis experimentally. Similar de-etiolation experiments could be conducted with a starchless mutant (e.g. lacking chloroplastic phosphoglucomutase, pgm). However, care would have to be taken since this mutation is also known to have secondary effects on lipid deposition in the developing seeds (Andriotis et al., Plant Physiol. 2012 Nov; 160(3): 1175–1186), which could indirectly affect the speed/extent of de-etiolation. We respectfully feel this is beyond the scope of the current manuscript.

“The surface area…” please define what is meant since membranes have two sides.

We selected and measured the stroma side of thylakoid membranes and clarify this in the text. The point is raised because the membrane is a bilayer, which is important when making comparisons between the measured membrane lipids quantification and measured membrane surface areas.

Subsection “Quantitative analysis of thylakoid surface area per chloroplast during de-etiolation” second paragraph: There is another study done in cell culture that has a similar design (Dubreuil et al. ), are the two studies compatible with each other in their conclusion and if not, what are the differences?

Although the two studies have a similar design, the study of Dubreuil et al. focused on transcript levels while we focus here on protein levels. However, we appreciate the reviewer’s question, since a comparison can be done. In the Discussion we now added a comparison between the transcriptome dynamics presented in Dubreuil et al. and our proteomic dynamics : “A significant change in the proteome was observed when comparing T24 and T0 but overall this change appeared gradual, indicating that increase of chloroplast associated proteins does not exactly follow the two-step induction of corresponding nuclear encoded transcripts reported previously (Dubreuil et al., 2018). “

“Our data suggest that the reorganisation of pre-existing molecules rather than de novo synthesis is responsible for the major chloroplast ultrastructural changes that occur between T0 and T4.”: This sentence is not perfectly clear to me. Maybe the authors can explain this a bit more in detail using examples.

We clarified our statement by explaining what data we compared: “We observed striking changes at the chloroplast ultrastructural levels, and in particular the formation of thylakoids between T0 and T4. However, our proteomic analysis indicated only a few changes in abundance of proteins between these time points, including proteins constituting the photosynthetic machinery (Figure 8—figure supplement 1 and Figure 5—source data 1). Also, we did not observe a significant increase in the major galactolipids constituting the lipid bilayer (Figure 7 and Figure 7—source data 1).”

Subsection “Chloroplast development: ‘Structure Establishment Phase’”: I think it is worth noting that the interactions between PSII complexes located in neighbouring thylakoid membranes trigger the stacking of the grana. It is therefore tempting to speculate that stroma lamellae are established first and that these membranes are then stacked after PSII complexes are inserted into the membrane because they provide the adhesion points between them.

Although several studies proposed that membrane stacking requires LHCII-PSII super complex (e.g. Albanese et al., 2020), the recent study by the Engel lab based on cryo electron tomography and mapping of photosynthesis related complexes challenges this idea (Wietrzynski et al., 2020). The structural basis and molecular mechanism underlying grana formation and stabilization is thus far from being elucidated and therefore we prefer not to speculate on this point.

Reviewer #2:This impressive manuscript describes a comprehensive, multifaceted analysis of the morphological and molecular changes that accompany photosynthetic establishment during seedling de-etiolation. Morphological data, focusing in particular on the photosynthetic thylakoid membranes, are derived using transmission electron microscopy (TEM), serial block face scanning electron microscopy (SBF-SEM), and confocal microscopy, while quantitative molecular data on the abundancies of proteins and lipids are derived using mass spectrometry and western blotting. The various data are acquired over a time course between 0 h and 96 h post illumination, and with a high level of temporal resolution. The data allow the authors to develop a mathematical model for the expansion of the surface area of thylakoids (reaching 500-times the surface area of the cotyledon leaf), which matches well with experimental observations from the SBF-SEM analysis for earlier, but not later, stages of de-etiolation. Moreover, the data point to a two-phase organization of the de-etiolation process, with the first phase ("Structure Establishment") characterized by thylakoid assembly and photosynthetic establishment, and the second phase ("Chloroplast Proliferation") characterized by chloroplast division and cell expansion.The data are of a high standard, and the depth and breadth of analysis in a single, unified study is unprecedented. While it is arguable that there are few major, completely novel insights reported here (indeed, in the Discussion, the authors very helpfully point out how many of the parameters they have measured are consistent with data reported elsewhere by others), this should not detract from the overall value of the study; a major and unique strength here is that all of the data have been acquired together and so are directly comparable. I have no doubt that this dataset will be extremely interesting to many researchers, and prove to be an invaluable resource for the plant science community. Consequently, I am sure that it will attract many citations.I have a few specific comments that I would like the authors to consider carefully, as follows.1) Figure 3. The 3D reconstructions are undoubtedly useful for deriving quantitative data, as they enabled the derivation of thylakoid surface area data to verify the mathematical model. However, it is very difficult to see anything clearly in the images shown in the figure. I wonder if the authors can make the images clearer, and then also point to and describe some of the key features. The videos do help a bit, but even these are not that clear.

We annotated the images in Figure 3, annotating the structures to which we refer in the text. We added a descriptive legend to the videos.

2) Subsection “Quantitative analysis of thylakoid surface area per chloroplast during de-etiolation”, second paragraph. It is here that the "two phases" model is first proposed. I really could not see a clear basis for proposing this model here, using the data that had been presented thus far. As I see it (and based on the way the two phases are described in the Discussion), one can't really propose this model until after the chloroplast number and cell size data have been presented.Moreover, the description of the second phase here ("and a second phase…") seems a bit inconsistent with the statement in the paragraph above that thylakoid surface area increases dramatically between T4 and T24, and much less between T24 and T96.

Indeed, the conclusions here were a bit premature and we restricted now the conclusion to emphasize only the fact that ultrastructural changes are mainly observed between T0 and T24. The text now reads: “Our quantitative observations confirmed that during chloroplast development the major ultrastructural changes (disappearance of prolamellar body, build-up of the thylakoids and their organization into grana) occurs within the first 24 hours of de-etiolation, and no drastic changes occur thereafter.”

3) Figure 6, and the related supplementary figure. Loading controls are missing here, and should be added.

We added actin as loading control on Figure 6. We did not add loading controls in the corresponding supplement because for this experiment, we quantified by calculating the mean of 3 replicates from independent experiments and normalized the data according to fresh weight. Therefore, we assumed that differences in loadings will be minimized by averaging and will contribute to the calculated standard deviation. The graphs corresponding to the quantification are now presented in (Figure 6—figure supplement 1 and we provide all raw data for quantifications in Figure 6—source data 1).

Also, it is stated that a number of proteins (PsbA, PsbD, PsbO, Lhcb2) are "detectable" at T0. To me, they look UNdetectable.

This was indeed a mistake that we corrected by replacing “T0” by “T4”.

4) Dividing chloroplasts. In subsection “Identification of a chloroplast division phase”, it is stated that the volume of dividing chloroplasts was measured, and we are referred to Figures 8E and 4B in support of this statement. However, it is not explained how this was done. More clear and specific explanation is needed. Was it the case that the authors sought out and measured dumbbell-shaped organelles, and quantified those? If so, images are needed to illustrate this point.

Chloroplasts in division were selected visually based on the presence of a constriction ring. We added this information in the text and illustrated it by examples of chloroplasts in division used for this analysis (3D reconstruction of one T24 and one T96 chloroplasts, Figure 8—figure supplement 1). The 3D volume of selected chloroplast was determined using the label analysis package of Amira software as indicated in the Materials and methods section.

And, I don't see anything relevant in Figure 4B – this callout apparently belongs in the following sentence. The statement that the average size of dividing chloroplasts was higher than that of all chloroplasts is not really surprising if the authors were measuring organelles just on the point of becoming two organelles.

We rephrased now this paragraph: “To test whether there is a correlation between chloroplast division and either volume or developmental stage, we measured the volume of dividing chloroplasts (selected visually based on the presence of a constriction ring, see Figure 8—figure supplement 1, Figure 8—source data 1) at T24 and T96 using images acquired by SBF-SEM. The average volume of dividing chloroplasts at T24 and T96 were consistently higher than the average volume of all chloroplasts (96 µm3 and 136 µm3 compared to 62 µm3 and 112 µm3, respectively) (Figure 4B, Figure 8E and Figure 8—source data 1) indicating that smaller chloroplasts at a specific time point are not dividing. This indicates that developing chloroplasts only divide once a certain chloroplast volume is reached. “ We also corrected the numbering of the label of this figure (J and K were missing).

5) Subsection “Model of thylakoid surface expansion over time”. The motivation for this section needs to be better introduced. When I first read it, I could not understand why the authors wished to again "determine the thylakoid membrane surface area", as this had already been discussed earlier in the manuscript.

For better clarity, we now rephrased the first sentence of this section as follows : “The quantitative molecular data for the major compounds of thylakoids (galactolipids and proteins) and estimation of chloroplast number per cell allowed us to mathematically determine the thylakoid membrane surface area per seedling and its expansion over time (molecular approach thereafter) and compare it to the surface estimated from the 3D reconstruction (morphometric approach thereafter).”

Also related to the modelling: Did the authors take into account the existence of appressed membranes when calculating the surface area exposed to the cytosol.

Yes, we calculated the surface corresponding to outer side of thylakoids. We reformulated the corresponding sentence to clarify it: “To calculate the surface area of outer membrane of thylakoids (i.e. surface exposed to the stroma in lamellae and facing the other thylakoid in appressed regions)”

And, assuming it is clearly established that there is a 1:1 relationship between these proteins and the relevant complexes, perhaps this should be stated and the relevant literature cited.

Done

Reviewer #3:This study by Pipitone et al. combines SBF-SEM microscopy with quantitative proteomics and lipidomics to explore chloroplast differentiation. Authors describe that chloroplast biogenesis occurs in a first phase of structure establishment with thylakoid biogenesis, followed by a second phase of chloroplast division. The images and 3D reconstructions are beautiful, the quantitative data are novel, and their integration offers a new perspective into the seedling de-etiolation process, a model system for physiological and molecular studies. However, in my opinion some aspects need to be better explained and significantly improved.The authors write: "After 8h of illumination (T8), we observed decreased abundance of only one protein (the photoreceptor cryptochrome 2, consistent with its photolabile property) and increased levels of only three proteins, which belonged to the chlorophyll a/b binding proteins category involved in photoprotection (AT1G44575 = PsbS; AT4G10340 = Lhcb5; AT1G15820 = Lhcb6"). This is striking, as many well studied proteins change in abundance during the first hours of de-etiolation. Actually, looking into the data set with the quantification data for of the ~5,000 proteins, it appears that many proteins do show significant changes between T0 and T8. For example PORA and ELIP, changes that are also reflected in Figure 6A.

We considered changes to be significant only if the q-value is below 0.01, and with this threshold value only the 4 above-mentioned proteins changed significantly in abundance during the first 8 hours. We now included in the text the analysis with less stringent statistic threshold value (qvalue < 0.05) and further clarified that we describe changes that are indicated significant by our statistic tests.

Related to the above, well known proteins for example phyA and HY5, that undergo drastic changes in abundance when etiolated seedlings are first exposed to light, do not show changes in T4,T8 and T12 relative to T0 in the proteomics data set. This raises questions about the proteomic approach (sensitivity of the method?) or the experimental setup. Could authors please comment on this? I feel that validation of the proteomics approach is critical, especially taking into account the central conclusion that "the first 12h of illumination saw very few significant changes in protein abundance".

We analysed phyA and HY5 levels by western blot, using the same protein samples used for the experiments presented in Figure 6 and its supplement (that were also prepared using the same conditions as for the proteomics). The data are now shown in Figure 6. We could observe a very fast decrease of phyA protein (starting already after 4 hours of illumination) and transient increased accumulation of HY5 that is consistent with previously published data (e.g. Debrieux et al. 2013, Hardtke et al. 2000, respectively). The observed fast degradation of phyA by western blot is not fully identical to the proteomic data, where we observed a significant phyA decrease not earlier than 72 hours of light exposure. We now propose explanations related to these differences as : “Overall the dynamics of the accumulation of proteins revealed by proteomics was similar to the dynamics observed by immunoblots (Figure 6 ; Figure 6—figure supplement 1 and Figure 5—source data 1), although not totally identical for some proteins (e.g. : phyA, HY5). The observed differences may be due to the detection methods of the two approaches (detection and relative quantification of individual peptides in proteomics versus detection of the full-length protein by immunoblot analysis) or other inherent limitations of proteomics when faced with low-abundance proteins like transcription factors. “

Subsection “Chloroplast development: ‘Structure Establishment Phase’” paragraph three: A reference is needed. Also, it is mentioned that PSII appears later than PSI, which does not seem to match the observation that PSII proteins appear earlier than PSI, or that the surface area occupied at early time points by PSII is greater than the one occupied by PSI. Please check.

That’s indeed a good point and we adapted the Discussion to mention this discrepancy between PSII/PSI ratio at early time points and timing of grana formation, and propose an alternative hypothesis: “however, it is intriguing to observe that PSII protein abundance is higher at early stages of thylakoid formation when grana have not yet been organised. Preferential localization of the PSI and PSII protein complexes in specific thylakoid membrane domains have been reported (lamellae and grana, respectively; (Wietrzynski et al., 2020)). Therefore, the timing of PSII/PSI relative abundance do not match with their preferential localization. It is possible that the formation of PSI still needs to be delayed until grana formation and PSII relocalization is initiated, which can prevent spillover between the two photosystems (Anderson, 1981)”

Are the calculations of thylakoid surface expansion over time consistent with previous available data using tomography? Please include.

One of the novelties of our work is that we quantified the thylakoid expansion over time of de-etiolation. To our knowledge, this has not been reported previously.

In the Introduction, authors could include mention of the massive transcriptional reprogramming that take place during de-etiolation. In addition, I think that comparison of the proteomics data with the transcriptomic changes during de-etiolation (well described in the literature) would allow further understanding of the distinct phases proposed. For the chloroplast proteins already present in the dark, how does this correlate with expression of the corresponding genes?

We thank the reviewer for this suggestion and we now added this information in the Introduction, which now read as : “ Photomorphogenic program is controlled by regulation of gene expression at different levels (Wu et al., 2014). Transcriptome analyses have revealed that upon light exposure, up to one third of Arabidopsis genes are differentially expressed, with 3/5 being are upregulated and 2/5 downregulated (Ma et al., 2001).”

We also now further compare our proteomic data with previously published transcriptomic data (see also the answer to the comment of reviewer 1) and added the correlation between mRNA and protein dynamics for some proteins, in the text which now reads : “At T96 the abundance of 607 proteins (12% of the identified) was increased which confirm the massive reorganization of the proteome following the reorganization of the transcriptome during photomorphogenesis (Ma et al., 2001). Proteins whose transcript levels decreases in response to light exposure were also downregulated at the protein levels (e.g. phyA and PORA) (Figure 6) (Ma et al., 2001). GO analysis combined with expression pattern–based hierarchical clustering highlighted that most photosynthesis-related proteins are globally coregulated (Figure 5—figure supplement 1, clusters 2 and 6) which correlates as well with the overall increase of their corresponding transcripts upon light exposure (Ma et al., 2001).”